# LASER: A Neuro-Symbolic Framework for Learning Spatio-Temporal Scene Graphs with Weak Supervision

**Jiani Huang**[§]    **Ziyang Li**[§]    **Mayur Naik**[§]    **Ser-Nam Lim**[†]

`{jianih, liby99, mhnaik}@seas.upenn.edu`    `sernam@ucf.edu`

[§]University of Pennsylvania    [†]University of Central Florida

## Abstract

Supervised approaches for learning spatio-temporal scene graphs (STSG) from video are greatly hindered due to their reliance on STSG-annotated videos, which are labor-intensive to construct at scale. Is it feasible to instead use readily available video captions as weak supervision? To address this question, we propose LASER, a neuro-symbolic framework to enable training STSG generators using only video captions. LASER employs large language models to first extract logical specifications with rich spatio-temporal semantic information from video captions. LASER then trains the underlying STSG generator to align the predicted STSG with the specification. The alignment algorithm overcomes the challenges of weak supervision by leveraging a differentiable symbolic reasoner and using a combination of contrastive, temporal, and semantics losses. The overall approach efficiently trains low-level perception models to extract a fine-grained STSG that conforms to the video caption. In doing so, it enables a novel methodology for learning STSGs without tedious annotations. We evaluate our method on three video datasets: OpenPVSG, 20BN, and MUGEN. Our approach demonstrates substantial improvements over *fully-supervised* baselines. On OpenPVSG, LASER achieves a unary predicate prediction accuracy of $27.78\%$ $(+12.65\%)$ and a binary recall@5 of $0.42$ $(+0.22)$. Furthermore, LASER exceeds baselines by $7\%$ on 20BN and $5.2\%$ on MUGEN in terms of overall predicate prediction accuracy.

## 1 Introduction

Understanding video semantics has gained prominence due to a wide range of applications such as video search, text-video retrieval, video question answering, video segmentation, and video captioning. Video semantics constitutes two crucial aspects: *spatial semantics*, which concern the entities in the video, their individual attributes, and their semantic relationships; and *temporal semantics*, which capture actions and properties evolving through time. For example, the video described in Figure 1 by the phrase *"pushing a box off the desk by hand"* involves entities like *"box"* and *"hand"*, which are connected by the spatial relation *"touching"*. It also features two temporally consecutive states: the *"box"* is first *"on"* the *"desk"*, and then *"not above"* the *"desk"*.

To explicitly learn combined spatial and temporal semantics, a structured representation called *Spatio-Temporal Scene Graph* (STSG) (Shang et al., 2017; Zhu et al., 2022) has been proposed to represent entity relations throughout a video. Existing approaches for learning STSG from video data are typically fully-supervised, e.g., Nag et al. (2023); Cong et al. (2021). They can potentially learn high-fidelity STSGs from video data but are greatly hindered in practice due to the complexity of low-level annotations that are laborious to obtain (Yang et al., 2023a).

Weak supervision emerges as a promising approach to address this challenge. For example, the vast availability of video captions provides a valuable source of weak supervisory signals. However, key difficulties arise in effectively learning STSGs from such weak supervision. Is it even feasible to use video captions given the sparsity and noise in the signals they provide? Captions often focus only on the primary objects, ignoring underlying details, and many temporal signals are either hidden or must be inferred. How can we provide useful fine-grained signals under such circumstances? To address these challenges, we propose transforming captions into *logical specifications* using large

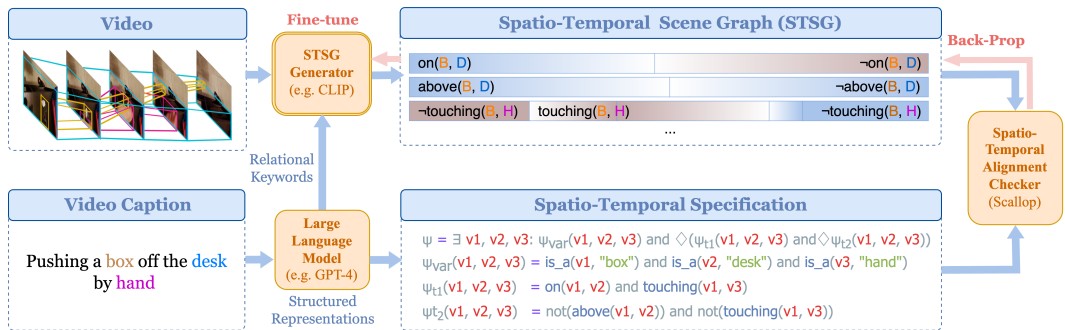

Figure 1: Illustration of the learning pipeline of LASER. The goal is to fine-tune a vision-language model to produce STSG without direct supervision on ground truth STSG labels. LASER relies on video captions for weak-supervision labels. We apply an LLM to extract a spatio-temporal specification from video captions. The LLM-inferred relational keywords, along with the input video, are then passed to a vision-language model to generate an STSG. At the end, a spatio-temporal alignment checker uses the specification to derive an alignment loss, capturing issues in the predicted STSG. The differentiable checker effectively back-propagates the loss to the vision-language model.

language models to explicitly reveal the hidden spatial and temporal information. This transformation creates a shared foundation to systematically align captions with predicted STSGs. The alignment process should a) capture both spatial and temporal nuances to provide fine-grained supervision for underlying STSG generators; b) allow diversity, naturalness, and fuzziness in the video and caption data; and c) account for common-sense knowledge that may be implicit or ambiguous in the captions.

We set out by designing STSL, a general and expressive Spatio-Temporal Specification Language for specifying fine-grained spatio-temporal properties. STSL is grounded in *Finite Linear Temporal Logic* ($LTL_f$) (De Giacomo & Vardi, 2013) which is used to describe temporal properties over finite traces of action and states. STSL subsumes action sequences commonly seen in video-action alignment tasks (Chang et al., 2019) while capturing additional temporal nuances such as "until" ($\mathbf{U}$) and "finally" ($\Diamond$). It also allows to express common-sense constraints for extra supervision. Finally, combined with relational predicates extracted from natural language, such as "is pushing off" and "lies above", it can even specify the open-domain spatial semantics of videos.

We now introduce LASER (Learning to Align for Spatio-tEmporal Representations), a novel framework to enable training STSG generators using only video captions. As illustrated in Figure 1, LASER enhances a vision-language model by aligning its predicted STSGs with STSL specifications derived from video captions using large language models. This alignment process is carried out in a divide-and-conquer fashion, where the caption is broken down into temporally related events, each of which must correspond with a portion of the STSG. We enhance the alignment process in two important aspects. First, to ensure precise optimization, we implement a neuro-symbolic alignment checker atop the Scallop framework (Li et al., 2023), making the alignment both probabilistic and differentiable. This enables seamless integration into an end-to-end learning pipeline. Second, to complement the weak supervision, we introduce a multi-faceted loss function that includes contrastive, temporal, and semantic components, which provide additional layers of supervision.

We conduct an extensive evaluation of LASER across various dimensions, demonstrating its broad utility and effectiveness. Being model-agnostic, LASER is capable of fine-tuning a wide range of STSG generators, including open-world generators in our evaluation such as CLIP (Radford et al., 2021), VIOLET (Fu et al., 2021), and SigLIP (Zhai et al., 2023), as well as closed-world generators like MLP classifiers for STSG generation. To evaluate its versatility, we apply LASER to diverse datasets such as OpenPVSG (Yang et al., 2023a), 20BN (Goyal et al., 2017), and MUGEN (Hayes et al., 2022), which encompass open-domain vocabularies, specifications with varied patterns, and both synthetic and complex real-world videos. Despite challenges such as an excessive number of relevant entities, diverse open-domain labels, and fuzzy captions, LASER consistently outperforms even fully supervised baselines in STSG generation on all three datasets. To further evaluate the data efficiency of LASER, we train the STSG generator with just $10\%$ of the training data, resulting in an average of $70.75\%$ of the performance gains achieved with the full dataset.

We summarize the main contributions of this work as follows:

1. We introduce a novel formulation of spatio-temporal scene graph learning as a weakly supervised task driven by video captions.
2. We design STSL, a general and expressive spatio-temporal specification language, for specifying fine-grained video semantics.
3. We implement a differentiable neuro-symbolic alignment checker to relate between an STSL specification and a spatio-temporal scene graph.
4. We propose LASER, a model-agnostic, end-to-end differentiable framework for learning spatio-temporal scene graphs with weak supervision from video captions.
5. We empirically evaluate LASER on three video understanding datasets, demonstrating superior performance in video semantics extraction tasks.

## 2 RELATED WORK

**Structured Representation of Image/Video Semantics.** Significant advances have been made in representing structured information within vision data. A widely adopted representation for capturing spatial semantics in images is *Scene Graph* (Kuznetsova et al., 2018; Lu et al., 2016), with various generation techniques emerging over the years (Zhu et al., 2022; Liu et al., 2021; Huang et al., 2020; 2021; Yang et al., 2018; Li et al., 2024). More recently, research has increasingly focused on extending these representations to integrate both spatial and temporal semantics in videos (Yang et al., 2023a; Li et al., 2022). However, learning spatio-temporal structures remains a challenging open problem, which LASER addresses by introducing a neuro-symbolic approach.

**Video Scene Graph Learning.** Learning video scene graphs has attracted significant attention from the vision community. Various tasks, including entity tracking, object identification, dynamic relation analysis, and pathfinding, are being explored (Sun et al., 2023; Xu et al., 2022a; Shang et al., 2017). New techniques involving spatio-temporal aware networks have been developed (Nag et al., 2023; Cong et al., 2021; Ji et al., 2021; Sun et al., 2019). A few recent works have developed point-solutions for extracting fine-grained video semantics (Lee et al., 2023; Apriceno et al., 2022; Chang et al., 2019). Such approaches includes both dynamic time warping (Dvornik et al., 2021; Chang et al., 2019; Richard et al., 2018; Ding & Xu, 2018), soft nearest neighbor (Han et al., 2022; Dwibedi et al., 2019), and semantic loss (Xu et al., 2022b). To our knowledge, LASER *is the first framework to train STSG generation models using video captions as weak-supervisory labels*.

**Vision Language Pretraining.** Vision language pretraining is crucial in video understanding and has a wide range of downstream applications. Current works have succeeded in learning visual representations using large-scale paired visual-textual data through contrastive learning in both image-text (Zhai et al., 2023; Radford et al., 2021; Jia et al., 2021) and video-text (Li et al., 2021; Xu et al., 2021; Miech et al., 2019) representation learning. Recent works also explore the viability of utilizing pretrained foundation models for generating image and video scene graphs (Shindo et al., 2024; Liang et al., 2024; Zhang et al., 2023; Yao et al., 2021).

## 3 METHODOLOGY

We begin by presenting the high-level problem definition. We are given a dataset $\mathcal{D}$ of video-caption pairs $(X, c)$, where $X = [x_1, \ldots, x_n]$ is a video containing $n$ frames, and $c$ is its video caption. We then transform the caption $c$ into a spatio-temporal specification $\psi$ in $\text{LTL}_f$ using a large language model. We wish to learn a neural model $M_\theta$ which extracts a spatio-temporal scene graph $\hat{\mathbf{r}} = M_\theta(X)$ that conforms to the corresponding specification $\psi$. During training time, given a loss function $\mathcal{L}$, we aim to minimize the following main objective:

$$J(\theta) = \tfrac{1}{|\mathcal{D}|} \sum_{(X,\psi) \in \mathcal{D}} \mathcal{L}(\Pr(\hat{\mathbf{r}} \models \psi \mid \theta), 1), \tag{1}$$

where $\Pr(\hat{\mathbf{r}} \models \psi \mid \theta)$ is the alignment score (probability of alignment) computed by our spatio-temporal alignment checker, conditioned on the model parameter $\theta$. We illustrate the full learning pipeline in Figure 1 and detail the process in this section. We first describe our video semantics representation using a probabilistic relational database (§3.1). Then, we present our specification language STSL (§3.2) and its alignment checker (§3.4). We further demonstrate how to automatically convert natural language captions to a specification in STSL using LLM (§3.3). In §3.5, we present our multi-faceted loss function design comprising contrastive, temporal, and semantic components.

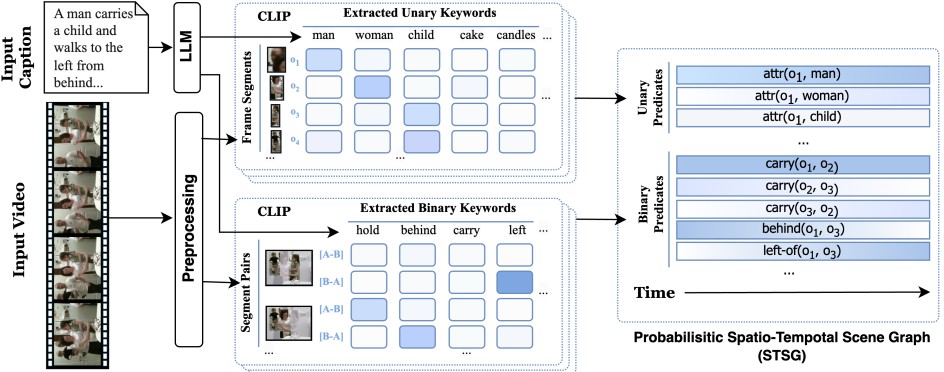

Figure 2: Pipeline illustration with CLIP as the backbone model for probabilistic STSG generation.

### 3.1 VIDEO TO PROBABILISTIC RELATIONAL DATABASE

A probabilistic spatio-temporal scene graph is a probabilistic relational database that contains two types of facts denoted by relations `unary_atom` and `binary_atom`, for unary and binary predicates respectively, each associated with a probability denoting the likelihood that the fact is true. For example, `0.05::unary_atom("deformed", 3, e)` means that *"entity `e` is unlikely to be deformed at time stamp 3,"* while `0.92::binary_atom("push", 10, h, b)` indicates that *"object `h` is highly likely to be pushing object `b` at time stamp 10."* This flexible representation supports the seamless incorporation of unary and binary keywords into the database. The unified probabilistic database enables LASER to be model-agnostic, supporting both closed-domain STSG classification models and open-world vision-language models for converting input video data into relational representations. With a unified formalization, an STSG generator, $M_\theta$, takes in pixel-based raw video data $X$, and generate a distribution of STSGs, which is encoded as a probabilistic relational database $\hat{\mathbf{r}}$.

We illustrate the STSG generation pipeline using an open-world vision-language model, CLIP, in Figure 2. The pipeline generates a probabilistic STSG from an input video and caption pair. An LLM first extracts relational keywords from the caption and passes the keywords to the STSG model. The video is then preprocessed by cropping objects in each frame and aligning them with unary keywords, while weighted masking is applied to object pairs to distinguish between subject, object, and background, aligning them with binary keywords. Consistent relationships, such as object categories, are aggregated across the video, resulting in the desired probabilistic STSG. For closed-domain STSG classification models, the pipeline remains the same in terms of video preprocessing. However, instead of relying on an LLM to extract keywords, these models predict directly over a predefined set of vocabularies. For brevity, throughout the paper, we use actual unary and binary keywords as predicate names, e.g. `deformed(2, e)` and `push(10, h, b)`.

### 3.2 SPATIO-TEMPORAL SPECIFICATION LANGUAGE (STSL)

Linear Temporal Logic (LTL) (Pnueli, 1977) is a formal logic system extending propositional logic with concepts about time. It is commonly used for formally describing temporal events, with applications in software verification (Chaki et al., 2005; Kesten et al., 1998) and control (Ding et al., 2014; Sadigh et al., 2014). As we operate on prerecorded, finite-length videos, our language is developed using $\text{LTL}_f$ (De Giacomo & Vardi, 2013), which supports LTL reasoning over finite traces. Thus, we use $\text{LTL}_f$ as a framework for specifying events and their temporal relationships.

Our STSL (Figures 4 and 5) further extends $\text{LTL}_f$ by introducing relational predicates and variables. It starts from the specification $\psi$ which existentially quantifies variables in an STSL formula. The formula $\varphi$ is inductively defined, with basic elements as relational atoms $\alpha$ of the form $a(t_1, \ldots, t_n)$. Note that the terms $\bar{t} = \{t_1, \ldots, t_n\}$ can contain quantified variables to be later grounded into concrete entities based on context $\Gamma$, noted by $\text{subst}_\Gamma(\bar{t})$. From here, $\varphi$ can be constructed using basic propositional logic components $\wedge$ (and), $\vee$ (or), and $\neg$ (not). The system additionally includes temporal unary operators $\square$ (always), $\lozenge$ (finally), $\bigcirc$ (next), and a binary operator $\mathbf{U}$ (until) (Albers et al., 2009). For example, the description "A hand continues to touch the box until it drops." can be

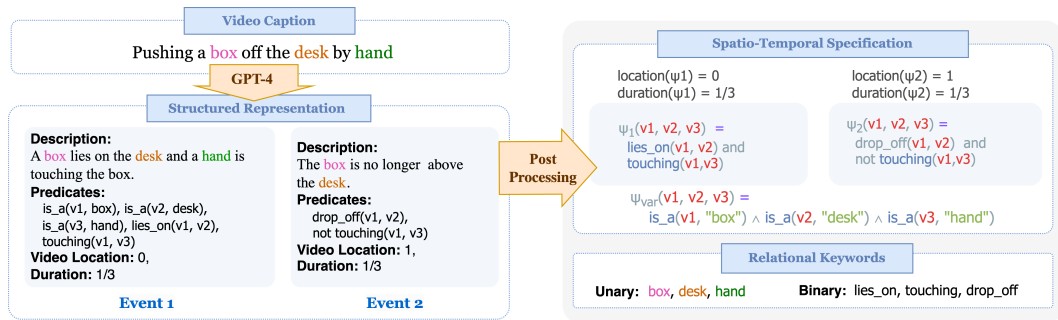

Figure 3: Pipeline utilizing 3-shot GPT-4 to convert natural language captions into: (1) programmatic spatio-temporal specification for alignment score calculation as input to the alignment checker, and (2) unary and binary keywords for predicting the probabilistic STSG as inputs to the neural model.

represented as an STSL formula

$$\psi = \texttt{touch}(\texttt{h, b}) \ \mathbf{U} \ \texttt{drop}(\texttt{b, \_}). \tag{2}$$

Note that an argument to the predicate `drop` is a wildcard (`_`), since we do not specify where does the box drops from. This formula might seem too strict since it requires the two events to be consecutive. To make the specification more natural, one can change the above formula to "$\Diamond(\texttt{touch(h, b)} \wedge \Diamond\texttt{drop(b, \_)})$". Here, the two events, `touch` and `drop`, need to happen in chronological order but are not required to be consecutive.

### 3.3 NATURAL LANGUAGE TO PROGRAMMATIC SPATIO-TEMPORAL SPECIFICATION

To leverage the abundance of video captions as weak supervision signals, we employ a large language model (LLM) to automatically extract a programmatic specification $\psi$ from each video caption $c$. Directly converting captions into a formal program is particularly challenging for an LLM, especially in a low-data language like STSL. We hence use a few-shot learning approach with an LLM to generate an intermediate structured representation of the caption in JSON format. For each caption $c$, our goal is to convert it into a series of events $\bar{e} = \{e_1, e_2, \ldots, e_n\}$. Each event includes (a) a detailed natural language description of the event, which guides the generation of subsequent details, (b) a series of unary, binary, positive, and negative predicates describing the semantics of the scenario, (c) the location of the event, $\text{loc}(e_i)$, in the video where the event occurs, represented as a fraction of the video length, and (d) the duration of the event, $\text{dur}(e_i)$, also expressed as a fraction. In Section 3.5, we explain how these structured representations are incorporated into the loss function.

To extract such structured representations from the caption, we designed a generic prompt template, which consists of the following components: (a) examples for temporal specification in fraction numbers: "0", "1/2", "2/3", "1". (b) scene graph keywords, such as object names and relations. (c) few-shot examples of caption and JSON structured representations pairs. We illustrate a caption and its structured representation in Figure 3, and the full prompt in the appendix.

The programmatic spatio-temporal specification is then generated by postprocessing the events in sequential order. Consequently, we can generate the programmatic spatio-temporal specification $\psi$ for the caption as a sequence of events in chronological order:

$$\psi = \Diamond_{e_i \in \bar{e}} \psi_i, \quad \psi_i = \bigwedge_{\phi_j \in \psi_i} \phi_j. \tag{3}$$

### 3.4 SPATIO-TEMPORAL ALIGNMENT CHECKING

Given a probabilistic database $\mathbf{r}$ that encodes a distribution of STSGs (§3.1), and a specification $\psi$ in STSL, we aim to measure the alignment score $\Pr(\mathbf{r} \models \psi)$ in an end-to-end and differentiable manner. Conceptually, each probabilistic fact $f$ in the database can be toggled on or off, resulting in $2^{|\mathbf{r}|}$ distinct *worlds*. Denoting each world (i.e. a discrete STSG) as $w \in \mathcal{P}(\mathbf{r})$, where $\mathcal{P}$ represents power-set, we can check whether the world $w$ satisfies the specification $\psi$ or not (Figure 5). From here, the alignment score can be computed as the sum of the probabilities of worlds satisfying $\psi$:

$$\Pr(\mathbf{r} \models \psi) = \sum_{w \in \mathcal{P}(\mathbf{r})} \Pr(w) \cdot \mathbb{1}[w \models \psi], \quad \Pr(w) = \prod_{f \in w} \Pr(f) \prod_{f' \in \mathbf{r} \setminus w} (1 - \Pr(f')) \tag{4}$$

(Formula) $\varphi ::= a(\bar{t}) \mid \varphi_1 \wedge \varphi_2 \mid \varphi_1 \vee \varphi_2 \mid \neg\varphi$
$\mid \bigcirc\varphi \mid \varphi_1 \mathbf{U} \varphi_2 \mid \Box\varphi \mid \Diamond\varphi$
(Specification) $\psi ::= \exists v_1, \ldots, v_k, \text{s.t. } \varphi$

Figure 4: The formal syntax of STSL. Here, $\wedge$, $\vee$, and $\neg$ represents logical "and", "or", and "not". Formula may also contain temporal operators $\bigcirc$ (next), $\mathbf{U}$ (until), $\Box$ (global), and $\Diamond$ (finally).

$$\begin{aligned}
\langle w, s \rangle &\models \psi & &\text{iff } \exists\Gamma, \langle\Gamma, w, s\rangle \models \varphi \\
\langle\Gamma, w, s\rangle &\models a(\bar{t}) & &\text{iff } a(\bar{c}) \in w[s] \wedge \bar{c} = \text{subst}_\Gamma(\bar{t}) \\
\langle\Gamma, w, s\rangle &\models \varphi_1 \wedge \varphi_2 & &\text{iff } \langle\Gamma, w, s\rangle \models \varphi_1 \wedge \\
& & &\quad \langle\Gamma, w, s\rangle \models \varphi_2 \\
\langle\Gamma, w, s\rangle &\models \neg\varphi & &\text{iff } \langle\Gamma, w, s\rangle \not\models \varphi \\
\langle\Gamma, w, s\rangle &\models \bigcirc\varphi & &\text{iff } \langle\Gamma, w, s+1\rangle \models \varphi \\
\langle\Gamma, w, s\rangle &\models \varphi_1\mathbf{U}\varphi_2 & &\text{iff } \exists i.s \leq i \wedge \langle\Gamma, w, i\rangle \models \varphi_2 \\
& & &\quad \forall k.s \leq k < i, \langle\Gamma, w, k\rangle \models \varphi_1
\end{aligned}$$

Figure 5: Formal semantics of STSL. $\langle w, s \rangle \models \psi$ means the STSL specification $\psi$ is *aligned* with the ST-SG $w$ starting from time $s$. We use $w \models \psi$ as an abbreviation for $\langle w, 1 \rangle \models \psi$.

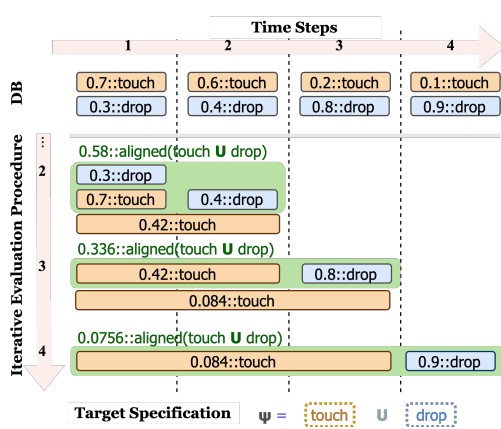

Figure 6: The evaluation process aligning a spatio-temporal scene graph (DB) with a specification `touch` $\mathbf{U}$ `drop`. This figure omits the predicate arguments, concentrating solely on matching sequential events.

Enumerating all possible worlds is intractable due to its exponential complexity. Existing general-purpose neuro-symbolic systems like Scallop (Li et al., 2023) employ scalable algorithms to approximate this probability and greatly reduce the probabilistic reasoning time. We also note that some of the STSG $w$ sampled from $\mathcal{P}(\mathbf{r})$ might be infeasible due to involving conflicting facts (e.g., a box is above and below a desk at the same time). To further enhance the logic deduction efficiency, we extend Scallop's "top-$k$ proofs" provenance to support general disjunctive constraints and early removal of infeasible STSGs that do not satisfy the specification.

LASER implements the alignment checker using our extended version of Scallop. It inductively computes the alignment between a temporal slice of $\mathbf{r}$ and an STSL formula. The whole specification $\psi$ is aligned if the full $\mathbf{r}$ satisfies $\psi$ with a concrete variable grounding $\Gamma$, which maps variables to concrete entities. We also extend Scallop with a variable assignment interface enumerating $\Gamma$ for constraint solving. We illustrate one simplified evaluation process in Figure 6. The checker iteratively aligns the predicted probabilistic events (simplified to just `touch` and `drop`) with the specification. At the 4th iteration, 4 different satisfying alignments are derived, yielding a final aggregated alignment score of 0.9916.

## 3.5 LOSS FUNCTION

**Contrastive Learning.** Unavoidable dataset biases exist in the specification. Contrastive learning can effectively reduce the bias and generate explanations of better quality. Let $(X_i, \psi_i)$ and $(X_j, \psi_j)$ be two datapoints in a mini-batch $B$, where $\psi_i$ and $\psi_j$ are the specifications for video $X_i$ and $X_j$ correspondingly. If $X_i \models \psi_j$, then it is an extra positive sample to the video $X_i$; otherwise, it is a negative sample to $X_i$. We can thus define our per-batch contrastive loss $\mathcal{L}_c(B)$:

$$\mathcal{L}_c(B) = \frac{1}{|B|^2} \sum_{(X_i, \psi_i) \in B} \sum_{(X_j, \psi_j) \in B} \mathcal{L}(Pr(M_\theta(X_i) \models \psi_j), \mathbb{1}[\psi_i = \psi_j]) \quad (5)$$

**Time-Span Supervision.** A video caption is expanded into a sequence of events using LLM, with each event assigned a specific temporal target, detailing its location and duration within the video, as illustrated in Section 3.3. By aligning the spatio-temporal specification $\psi$ with the video, we can identify when its sub-specifications, $\psi_1, \psi_2, \ldots, \psi_n$, are met. This alignment facilitates weak supervision across the entire time span. We define $\sigma(s, l, d) \in [0, 1]$, the time span alignment score, as a function on actual time stamp $s$, expected time stamp $l$, and expected event duration $d$. In particular, $\sigma(s, l, d)$ should peak at 1 when the event happens exactly at the expected locations ($s = l$). In practice, we embed $\sigma$ into the computation of probabilistic alignment between STSG $w$ and an

| Method (with LASER) | | Unary | | | Binary | | |
|---|---|---|---|---|---|---|---|
| | | R@1 | R@5 | R@10 | R@1 | R@5 | R@10 |
| VIOLET | Base | 0.0660 | 0.1855 | 0.2983 | 0.0460 | 0.1307 | 0.2636 |
| | Fine-tuned | 0.0878 | 0.2574 | 0.3463 | 0.0501 | 0.2028 | 0.3451 |
| | Incr. | ↑ 0.0218 | ↑ 0.0719 | ↑ 0.0480 | ↑ 0.0041 | ↑ 0.0721 | ↑ 0.0815 |
| SigLIP | Base | 0.0000 | 0.0179 | 0.0483 | 0.0000 | 0.0362 | 0.1667 |
| | Fine-tuned | 0.1467 | 0.2627 | 0.3152 | 0.0347 | 0.1624 | 0.3012 |
| | Incr. | ↑ 0.1467 | ↑ 0.2448 | ↑ 0.2669 | ↑ 0.0347 | ↑ 0.1262 | ↑ 0.1345 |
| CLIP | Base | 0.1633 | 0.3381 | 0.4404 | 0.0197 | 0.0673 | 0.0988 |
| | Fine-tuned | 0.2778 | 0.5231 | 0.6402 | 0.1482 | 0.4214 | 0.5398 |
| | Incr. | ↑ 0.1145 | ↑ 0.1850 | ↑ 0.1998 | ↑ 0.1284 | ↑ 0.3540 | ↑ 0.4410 |

Table 1: We show the performance improvements of base backbone models and their fine-tuned version, on the R@k metrics of unary and binary predicate prediction. As shown by the increments, LASER's weak supervisory learning framework significantly enhances all three models' performance on the STSG extraction tasks.

atomic specification $a(\bar{t})$, where we utilize the expected location $\text{loc}(a)$ and the duration $\text{dur}(a)$:

$$\sigma(s, l, d) = \max(0, 1 - \frac{2|s-l|}{d}), \tag{6}$$

$$\Pr(\langle \Gamma, w, s \rangle \models a(\bar{t})) = \Pr(a(\bar{c}) \mid a(\bar{c}) \in w[s] \wedge \bar{c} = \text{subst}_\Gamma(\bar{t})) \cdot \sigma(s, \text{loc}(a), \text{dur}(a)). \tag{7}$$

**Semantic Loss.** To provide further supervision, we resort to human knowledge encoded in the form of integrity constraints. We introduce semantic loss reflecting the probability of violating the integrity constraints. For example, an entity in a video cannot be `open` and `closed` at the same time; an entity that is not `bendable` cannot be `deformed`. These integrity constraints may interweave so heavily that it is hard to use a simple disjoint multi-class classifier to enforce. We encode all integrity constraints in the form of first-order logic rules, and our reasoning engine generates the probability that these constraints are violated. We thus have the per-sample semantic loss $\mathcal{L}_s(X_i)$ as an extra-weighted term after calculating the other loss components. Let $n$ be the number of integrity constraints and let $\text{IC}_i$ be the $i$-th integrity constraint, we have

$$\mathcal{L}_s(X_i) = \sum_{i=1}^{n} \mathcal{L}(\Pr(M_\theta(X_i) \not\models \psi_{\text{IC}_i}), 0). \tag{8}$$

## 4 EVALUATION

Our evaluation attempts to answer several key questions about LASER. How effective is LASER in learning STSG generators? How does its performance compare to fully supervised baselines and existing methods? Is STSL versatile and expressive enough when applied to diverse specification patterns? Finally, how essential is the multifaceted design of LASER's loss function?

To address these issues, we evaluate LASER on three datasets: OpenPVSG (Yang et al., 2023a), a realistic dataset with diverse and fine-grained STSG annotations, 20BN (Goyal et al., 2017), a video dataset focusing on daily actions, and MUGEN (Hayes et al., 2022), a synthetic dataset containing gameplay footages. These datasets vary significantly in their temporal patterns, showcasing the versatility of STSL. Specifically, OpenPVSG captions focus on natural and complex events, 20BN captions are in the form of action pre-conditions and post-conditions, while MUGEN captions describe consecutive action sequences performed by the main protagonist in the game.

Our main result shows significant improvements of LASER over fully supervised baselines. On OpenPVSG, LASER achieves a unary predicate prediction accuracy of 27.78% and a binary recall@5 of 0.42, surpassing the best fully supervised baseline by 12.65% and 0.22, respectively. Furthermore, LASER outperforms baselines by 7% on 20BN and 5.2% on MUGEN in overall predicate prediction accuracy. We now delve into the experiments conducted on each of these datasets.

### 4.1 OPENPVSG DATASET

**Dataset.** The OpenPVSG (Yang et al., 2023a) dataset comprises 400 videos sourced from Ego4D (Grauman et al., 2021), VidOr (Shang et al., 2019; Thomee et al., 2016), and EpicKitchen (Damen

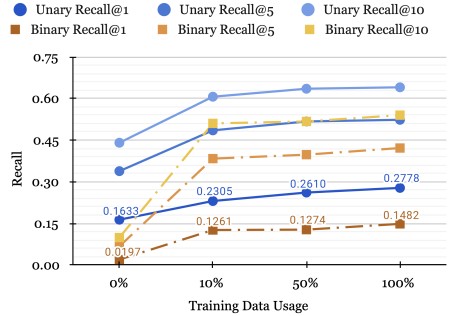

Figure 7: Data-efficient fine-tuning on OpenPVSG dataset with LASER: Providing only 10%, 50%, and 100% of the training dataset significantly enhances the performance of CLIP model.

| Method | Unary | Binary | | |
|---|---|---|---|---|
| | Acc. (%) | R@1 | R@5 | R@10 |
| IPS-Vanilla | 15.13 | 0.0741 | 0.1081 | 0.1109 |
| IPS-Filter | 13.14 | 0.0777 | 0.1040 | 0.1133 |
| IPS-Conv | 15.13 | 0.0861 | 0.1143 | 0.1218 |
| IPS-Trans | 14.67 | 0.1419 | 0.2032 | 0.2207 |
| VPS-Vanilla | 5.49 | 0.0374 | 0.0517 | 0.0531 |
| VPS-Filter | 5.46 | 0.0405 | 0.0480 | 0.0488 |
| VPS-Conv | 7.46 | **0.1616** | 0.1781 | 0.2343 |
| VPS-Trans | 5.46 | 0.1019 | 0.1499 | 0.1562 |
| LASER-CLIP | **27.78** | 0.1482 | **0.4214** | **0.5398** |

Table 2: Comparison between *weakly supervised* LASER-CLIP and *fully supervised* IPS and VPS methods on various backbones trained on the full OpenPVSG. LASER-CLIP significantly outperforms all baselines, except on Binary R@1, despite using *weak supervision*.

et al., 2022; 2018). This dataset offers fine-grained ground truth annotations of STSGs for $150K$ frames, encompassing 126 object classes and 57 relation classes. We train on $1,832$ video-caption pairs, and evaluate on 438 video-STSG pairs.

**Experimental Setup.** As illustrated in Figure 2, our objective is to train an STSG generator capable of taking a video clip with object bounding boxes as input and predicting the properties, attributes, and relationships between objects. We leverage LASER to fine-tune three vision-language models—VIOLET (Fu et al., 2021), SigLIP (Zhai et al., 2023), and CLIP (Radford et al., 2021)—using weak supervision from captions. These models predict both similarity scores between cropped objects and unary predicate keywords, as well as between object pairs and binary predicate keywords, resulting in a probabilistic STSG. All backbone models support open-world vocabularies and are thus robust to the fuzziness present in GPT-4-generated structured representations.

**Evaluation Metric.** We evaluate model performance using Recall@$k$ (R@$k$) which estimates whether the ground truth label is within the top-$k$ prediction of a given model. During evaluation, the model processes (a) the full vocabulary of object and relation classes and (b) preprocessed cropped objects and object pairs, predicting the probabilistic STSG. In particular, unary R@$k$ assesses object category prediction capability, while binary R@$k$ evaluates pair-wise prediction of binary relations.

**Backbone models significantly improve after been weakly supervised by LASER.** We validate LASER' effectiveness in learning STSGs with weak supervision by comparing the performance improvements of the backbone models after fine-tuning. As shown in Table 1, LASER significantly enhances backbone performance using only captions for weak supervision on the OpenPVSG dataset.

**Data efficiency of LASER.** To further assess LASER's data efficiency, we train the model on 10% and 50% of the training dataset. As illustrated in Figure 7, even with just 10% of the training data (183 video-caption pairs), LASER significantly enhances the unary R@1 from 0.1633 to 0.2305 and the binary R@1 from 0.0197 to 0.1261. On average, using just 10% of the data achieves 70.75% of the performance obtained with the full dataset, highlighting LASER's data efficiency.

**Weak-supervision may outperform full-supervision.** To better understand the efficacy of weak supervision, we also compare them against fully supervised methods. We study 8 fully supervised baselines, which employ two different video panoptic segmentation strategies: Image Panoptic Segmentation with Tracker (IPS) and Video Panoptic Segmentation (VPS). For relation extraction, the baselines employ 4 model architectures: (1) Vanilla: fully-connected layers, (2) Filter: handcrafted filters, (3) Conv: 1D-convolutional layers, and (4) Trans: transformer encoders. As shown in Table 2, LASER significantly outperforms the best fully supervised methods in all metrics except for binary R@1, where it ranks just after the top-performing VPS-Conv.

## 4.2 20BN DATASET

**Dataset.** The 20BN dataset consists of (a) video and action pairs of humans performing everyday actions with ordinary objects (Goyal et al., 2017), (b) expert designed pre-conditions and post-

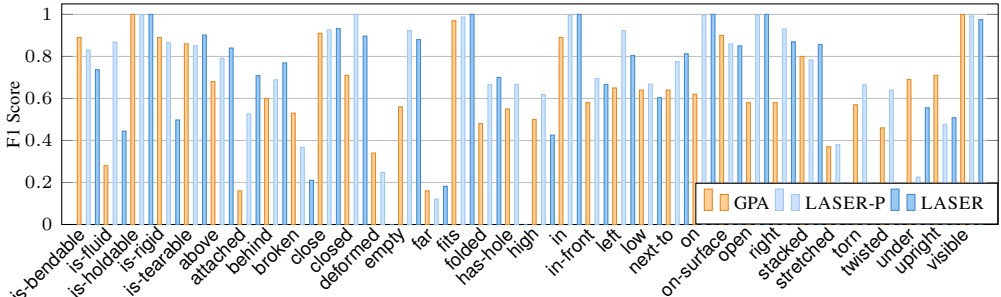

Figure 8: Per-predicate F1 score performance comparison of LASER, LASER-P, and GPA, all trained on the full 20BN dataset. LASER-P outperforms GPA on 71% of predicates, and LASER outperforms GPA on 59% of the predicates.

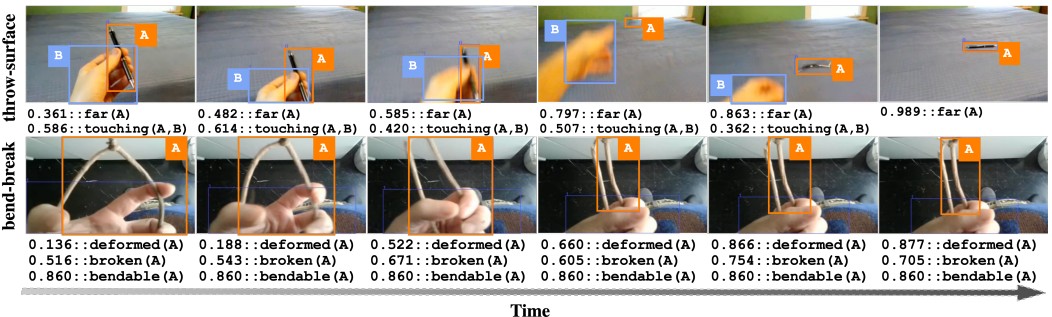

Figure 9: Qualitative study of the model trained with LASER on the full 20BN dataset. Each row displays a sequence of frames from a video, with bounding boxes labeled by object IDs. The left side of each row shows the action label, while the bottom of each row lists the attributes and relationships associated with the objects, along with the corresponding likelihoods of these facts holding true.

conditions for the actions in the PDDL language (Migimatsu & Bohg, 2022), and (c) frame-based object bounding boxes (Materzynska et al., 2020). There are 172 actions with 37 underlying predicates in this dataset, capturing object attributes, object states, and relationships between two objects. Specifically, there are 6 static predicates, 21 unary predicates, and 10 binary predicates. We train on $10,000$ training datapoints and test on $14,816$ data points.

**Experimental Setup.** In the 20BN dataset, each video is assigned an action label from a set of 172 possible actions. Each action is annotated with a natural language description and a logical specification, represented as a pair of pre-condition and post-condition in PDDL format. These PDDL specifications naturally map to the $\Diamond(\psi_{\text{pre}} \wedge \Diamond\psi_{\text{post}})$ structure in STSL. This setup enables us to assess the performance difference between caption-generated specifications and ground truth programmatic specifications. LASER received the generated specifications from natural language captions using GPT-4 as weak supervision label. For ablation study, we refer to a variant of our method that uses ground truth specifications as weak supervision labels as LASER-P.

**Evaluation Metrics.** We consider exact accuracy and F1-score as the metrics to evaluate STSG generators for 20BN dataset. Specifically, we compute F1-score and accuracy for each predicate in the vocabulary, and evaluate the weighted average of F1-score and accuracy for overall evaluation.

**Model Architectures.** Our backbone STSG generator model comprises an S3D (Xie et al., 2018) video encoding model pretrained on Kinetics 400 (Kay et al., 2017), followed by ROIpooler for extract object embeddings, then passed to MLP layers for relation classification.

**Quantitative and Qualitative Study.** We compare our approach against GPA (Migimatsu & Bohg, 2022), a weakly supervised baseline which uses the ground truth specification as learning signals. Our evaluation shows that LASER-P, with the same programmatic supervision, achieves a higher average F1-score of $0.77$ and accuracy of $91\%$, compared to the baseline's $0.74$ F1-score and $76\%$ accuracy. LASER, using only natural language descriptions, achieves a comparable F1-score of $0.73$ and better accuracy of $83\%$. Furthermore, LASER-P outperforms the baseline in $71\%$ of the

| Setup | Contr. | Sem. | Tmp. | F1 |
|---|---|---|---|---|
| 1 | ✓ | | | 0.31 |
| 2 | ✓ | ✓ | | 0.31 |
| 3 | ✓ | | ✓ | 0.58 |
| Full | ✓ | ✓ | ✓ | **0.77** |

| #Data | LASER | VT-TWINS | TempCLR | Supervised |
|---|---|---|---|---|
| 100 | **46.0** | 37.0 | 16.4 | 52.0 |
| 1000 | **48.5** | 22.8 | 17.10 | 52.0 |
| 5000 | **59.6** | 55.6 | 34.91 | 54.4 |

Table 3: Ablation study on loss function components (contrastive, semantic, and temporal losses), trained and evaluated on the full 20BN dataset. By ablating on loss function components, we find all are essential to the performance of our method.

Table 4: Data efficiency study of the LASER model trained on the MUGEN dataset, compared against VT-twins, TEMP-CLR, and fully supervised baselines. The models were trained on 100, 1,000, and 5,000 data points from the MUGEN dataset, evaluated using action prediction accuracy (%).

fine-grained predicate recognition tasks, and LASER outperforms 59%, as shown in Figure 8. For the qualitative study, we present the STSG generator's frame-wise predictions on several test data points in Figure 9. In addition to this, we conduct an ablation study, detailed in Table 3, to evaluate the impact of different components of our loss function design.

### 4.3 MUGEN DATASET

**Dataset.** MUGEN (Hayes et al., 2022) is a synthetic dataset that is based on an open-sourced platform game CoinRun (Cobbe et al., 2019). Each datapoint contains a 3.2s video snippet of the game-play and a corresponding automatically generated text describing the video. The agent in the game can perform 6 actions: `walk`, `jump`, `kill`, `collect`, `die`, and `climb`. A video may contain any sequence of actions, which may be simple or complex. We train on $5,000$ training datapoints and test on $12,851$ datapoints.

**Experimental Setup.** In the MUGEN dataset, each video is paired with a natural language description that can be represented as an ordered list of actions $[a_1, a_2, \ldots, a_n]$. The corresponding STSL specifications are naturally expressed in the logical form $a_1 \mathbf{U} a_2 \ldots \mathbf{U} a_n$. LASER uses these generated specifications, derived from natural language captions via a crafted semantic parser, as weak supervision labels. We evaluate the model's performance based on action prediction accuracy.

**Model Architectures.** Since the MUGEN videos differ significantly from natural videos, our backbone model is trained from scratch. Our backbone STSG generator model comprises an S3D video encoding model pretrained on Kinetics 400, followed by ROIpooler for extract object embeddings, then passed to MLP layers for relation classification.

**Quantitative Study.** We evaluate our action prediction performance on MUGEN and compare it to caption supervised VT-TWINS (Ko et al., 2022), TempCLR (Yang et al., 2023b), and a directly supervised baseline using the same model architecture as ours but with ground truth labels (Supervised). We provide all baselines with extra annotations on the start- and end-frame of each clip-based textual description. As shown in Table 4, LASER has better action prediction accuracy despite receiving less supervision than the baselines. Moreover, our weakly supervised model even achieves better accuracy than the Supervised method on when trained on $5000$ data points.

## 5 CONCLUSION, LIMITATION, AND FUTURE OUTLOOK

We propose LASER, a model-agnostic, end-to-end differentiable framework for learning spatio-temporal scene graphs with weak supervision from video captions. We evaluate our work on three datasets OpenPVSG, 20BN, and MUGEN which exhibit diverse temporal properties. LASER significantly outperforms even the fully supervised baselines on STSG generation tasks.

**Limitations.** LASER still faces scalability challenges regarding both video duration and the number of objects present in the video. Learning STSGs over long time horizons with weak signals remains an open problem. Additionally, LASER's performance is limited by the quality of the video captions. It is unclear whether more specifications can be elicited from LLMs to reduce reliance solely on captions. We aim to explore these directions in future work.

ACKNOWLEDGEMENTS

We sincerely thank the anonymous reviewers for their valuable feedback and constructive suggestions. We also gratefully acknowledge Mayank Keoliya, Matthew Kuo, Amish Sethi, and Neelay Velingker for their help on the project. This research was supported by ARPA-H program on Safe and Explainable AI under the award D24AC00253-00 and NSF award CCF 2313010.

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

## A  NATURAL LANGUAGE TO STSL SPECIFICATION

We designed a generic prompt template to convert a caption into its structured representation. The prompt template consists of the following components:

1. Set the LLM to be "a super user in logic programming".
2. Define video location and duration as fractions of the whole video length.
3. Include desired scene graph keywords, such as object names and relations.
4. Provide n-examples of captions and their corresponding JSON structured representations.

We include the full prompts for OpenPVSG dataset and 20BN-Something-Something dataset here.

### A.1  CHARACTER SETUP

We setup the agent character for all the datasets with this prompt.

```
You are a super user in logic programming.
```

### A.2  CONSTANT DEFINITION

We setup the examples for temporal specification for all datasets.

```
Examples of precise video locations: 1/4, 1/2, 2/3, 1.
Examples of event durations: 1/4, 2/3, 1.
```

We give the concept of "unary predicate" and "binary predicate" for all datasets.

```
A static predicate represents the property of the target object.
For example, is_tearable(A) means the A can be teared.
A unary predicate takes in one argument.
For example, close(A) means A is close to the camera.
A binary predicate takes in two arguments.
For example, above(A, B) means A is above B.
```

### A.3  SCENE GRAPH VOCABULARIES

To bias the GPT-4 model to generate more related words align with the desired scene graph vocabulary, we added the OpenPVSG vocabularies into the prompt as well.

We show the scene graph vocabulary part of OpenPVSG prompt below.

```
The entites in the video can be:
    adult, baby, bag, ..., tree, wall, water
The relations in the video can be:
    beside, biting,  blowing, brushing, ..., watering, wearing
```

We show the scene graph vocabulary part of 20BN-Something-Something prompt below.

```
Static predicates:
    is_bendable, is_fluid, is_holdable, is_rigid, is_tearable, neq
Unary predicates:
    broken, close, closed, deformed, empty, far, ..., twisted, upright
Binary predicates:
    above, attached, behind, fits, in, ..., touching, under, visible
Constants:
    hand
```

### A.4  FEW-SHOT EXAMPLE

In both cases, we provide the prompt with 3 examples converting the caption into a structured representation with json format. The target json file includes a detailed version of caption, the programmatic version in logical predicates, video location of the event and the duration of the event.

We show one example mapping the caption into json format below for OpenPVSG dataset below.

```
Caption: "The young boy receives another gift and sits on the floor."
Action json:
{
    "caption": "The young boy receives another gift and sits on the floor.",
    "sequential descriptions": [
        "boy A receives gift B",
        "boy A sits on the floor C",
        ],
    "time stamps": {
        "1": {
            "decription": [
                "boy A receives gift B",
            ],
            "programmatic": [
                "holding(B, D)",
                "name(A, boy)",
                "name(B, gift)",
            ],
            "duration": "1/2",
            "video location": "0"
        },
        "2": {
            "decription": [
                "boy A sits on the floor C",
            ],
            "programmatic": [
                "sitting on(A, C)",
                "name(A, boy)",
                "name(C, floor)",
            ],
            "duration": "1/3",
            "video location": "1"
        }
    }
}
```

We show one example mapping the caption into json format below for 20BN dataset below.

```
Action: dig
Action predicate: dig(A, B)
Action description: Digging [A] out of [B].
Action json:
{
    "action": "dig",
    "action_pred": "dig(A, B)",
    "explanation": "Digging [A] out of [B]",
    "static properties": [
        "A is holdable",
        "B is not rigid",
        "entity A and B are not equvalent"
    ],
    "programmatic version": [
        "is_holdable(A)",
        "not(is_rigid(B))",
        "neq(A, B)"
    ],
    "time stamps": {
        "1": {
            "decription": [
                "Not A and B are far away from the camera",
                "A is in B",
                "A and B are not touched by hand",
                "A is not visible, but B is visible,
                and there is a hand that is visible"
            ],
            "programmatic": [
                "not(far(A)), not(far(B))",
                "in(A, B)",
                "not(touching(A, hand)), not(touching(B, hand))",
                "not(visible(A)), visible(B), visible(hand)"
            ],
            "duration": "1/5",
            "video location": "0"
        },
        "2": {
            "decription": [
                "A become visible",
                "A is in a hand",
                "A is not touching B"
```

```
        ],

        "programmatic": [
            "visible(A)",
            "in(A, hand)",
            "not(touching(A, B))"
        ],
        "duration": "1/5",
        "video location": "1"
    }
  },
}
```

We have implemented corresponding post-processors to parse the json structures into STSL programs.

## B    EXPERIMENTAL DETAILS

### B.1    HARDWARE

All experiments are conducted on two machines:

1. two 20-core Intel Xeon CPUs, four GeForce RTX 2080 Ti GPUs, and 768 GB RAM,
2. 3.00GHz Intel Xeon machine with 48 CPUs, 8 A100s, and 1.5T RAM.

### B.2    OPENPVSG

**Dataset.** The OpenPVSG Yang et al. (2023a) dataset comprises 400 videos sourced from Ego4DGrauman et al. (2021), VidOrShang et al. (2019); Thomee et al. (2016), and EpicKitchenDamen et al. (2022; 2018). This dataset offers fine-grained ground truth annotations of STSGs for $150K$ frames. These scene graphs encompass 126 object classes and 57 relation classes, including pixel-level object masks. Each video is accompanied by multiple captions with their corresponding video snippets. The training dataset contains $1,538$ caption-video pairs, while the testing dataset includes 295 caption-video pairs. We train and test on these caption-video snippet pairs.

**Experimental Setup.** As illustrated in Figure 2, our objective is to train an STSG extraction model capable of taking a video clip along with object bounding boxes as input and predicting the properties, attributes of objects, and relationships between objects. As the ground truth programmatic specification is not provided in the OpenPVSG dataset, we only adopt the LASER-**C** architecture. The 3-shot GPT-4 model processes the captions and converts them into structured representations of events with logical predicates $\psi_i$. By postprocessing, we extract all the unary and binary keywords from the GPT-4 generated $\psi$, and use them as input to the STSG extraction model. We finetune the SigLIP model to generate the probabilistic spatio-temporal scene graph. This model predicts both the similarity scores between the cropped objects and the unary predicate keywords, and the similarity scores between object pairs and the binary predicate keywords, yielding a probabilistic STSG. Due to SigLIP's characteristic as an embedding-based model, it supports open-world vocabulary and can handle the fuzziness that may be present in the GPT-4 generated structured representations. Due to the hardware limitation, we set the batch size to 1, and our contrastive loss is obtained by randomly sampling another negative specification other than the current specification.

**Evaluation Metrics.** We use recall@k for evaluating the unary and binary STSG generating performance. The recall metric checks given the ground truth bounding boxes and object pairs, whether the predicted unary predicate, in this case, object names, ranks within the top-k among the ground truth unary predicates. Then we compare the accuracy against the ground truth binary labels.

To obtain the evaluation results for the OpenPVSG baselines, the object identification results need to be isolated from the entire prediction pipeline. The IPS-based OpenPVSG models first predict a class label for each pixel, then aggregate similar predictions into a single object using Unitrack. In contrast, the VPS-based models directly predict object tubes with corresponding labels. To separate object identification from the segmentation task, we identify predicted objects that sufficiently overlap with ground truth objects (with an IOU greater than 0.5). We then calculate the average label for each predicted object across the entire video. The label probability is determined by the proportion of frames in which the object is assigned a particular label, relative to the total frames in which the object appears.

**Learning Setup** For STSG generator, we finetune the "google/siglip-base-patch16-224", "openai/clip-vit-large-patch14", and the pretrained violet model checkpoint with learning rate 0.000001 for 10 epoches. To ensure the inference efficiency, we adopt sampling technique, uniformly sample the whole video frames, and ensure the final generated video has at most 12 frames. Further, we preserve only 10 largest objects on the image, and for each frame, we sample 100 relation pairs at training time, and 300 relation pairs at test time. We set the learning batch size to 1.

### B.3 LLaVA-Video Zero-Shot Transferability

We trained CLIP model with LASER method on 10K LLaVA-Video dataset, and test on OpenPVSG evaluation dataset, showing generalizability, robustness, and scalability.

### B.3.1 Dataset Preprocessing with LLM + Compiler Pipeline

**OpenPVSG Caption to STSL Program** The OpenPVSG captions average 12.69 words, containing an average of 1.69 events in the GPT-extracted representations. The failure rate for converting captions into valid executable STSL formulas is 0.78%. A total of 453 unique unary keywords and 679 unique binary keywords were identified, covering: 92.86% ground-truth unary and 94.74% ground-truth binary keywords in the OpenPVSG dataset. 55.56% ground-truth unary and 61.54% ground-truth binary keywords in the Action Genome dataset. 20% ground-truth unary and 12.88% ground-truth binary keywords in the VidVRD dataset. Notably, these closed-world datasets include keywords like "unsure," "other relation," and "not contacting," which rarely occur in natural scenes.

**LLaVA-Video 10K Dataset Caption to STSL Program** To demonstrate the capability of our LLM + compiler pipeline in handling long and complex caption descriptions, we evaluated it on LLaVA-Video, a dataset with detailed captions released on October 4, 2024. We study the quality of the extracted captions on a randomly sampled subset of 10,000 video clips (each under 30 seconds). The LLaVA-Video 10K captions average 233.05 words, and the pipeline extracted 4.03 events per video on average. The failure rate for converting captions into valid executable STSL formulas was 0%. A total of 18,458 unique unary keywords and 4,492 unique binary keywords were identified, covering: 91.27% ground-truth unary and 89.47% ground-truth binary keywords in the OpenPVSG dataset. 80.56% ground-truth unary and 69.23% ground-truth binary keywords in the Action Genome dataset. 82.86% ground-truth unary and 31.82% ground-truth binary keywords in the VidVRD dataset. Notably, 91.99% of LLaVA-Video samples originate from YouTube, with the remaining samples sourced from ActivityNet, Charades, Ego4D, NextQA, and YouCook2. This highlights a significant domain shift between the LLaVA-Video dataset and all evaluation datasets.

### B.3.2 Keyword Analysis

We present the top 10 most frequent unary and binary keywords extracted from captions in both datasets to further illustrate the diversity and robustness of our method.

| LLaVA-Video Unary | | LLaVA-Video Binary | | OpenPVSG Unary | | OpenPVSG Binary | |
|---|---|---|---|---|---|---|---|
| Category | Count | Category | Count | Category | Count | Category | Count |
| women | 1020 | hold | 2531 | adult | 823 | on | 264 |
| hand | 886 | wear | 2275 | child | 569 | holding | 211 |
| man | 863 | on | 1103 | man | 322 | picking | 163 |
| text | 499 | in | 811 | ball | 254 | placing | 148 |
| child | 421 | with | 725 | I | 206 | using | 135 |
| camera | 419 | adjust | 386 | dog | 187 | toward | 113 |
| character | 414 | color | 362 | toy | 127 | in | 90 |
| hands | 256 | stand | 346 | woman | 111 | throwing | 83 |
| room | 243 | near | 286 | Baby | 106 | playing | 66 |
| object | 239 | sit | 275 | camera | 104 | sitting on | 65 |

Table 5: Comparison of LLaVA-Video and OpenPVSG categories and their counts.

### B.3.3 Cost Analysis

Cost of Generating the STSL from Intermediate Representations Generating all intermediate representations with GPT for the $10K$ dataset costs approximately $50 and takes around 15 minutes using parallelization techniques. The compilation process from the $10K$ GPT-generated structured representations to executable STSL programs requires about 1 minute.

### B.4 Zero-shot Tranferability

Training with Noisy Object Trajectories and Long Captions on LLaVA-Video As the LLaVA-Video dataset does not provide ground truth object mask-level trajectories. To address this, we preprocess the videos using SAM2.1 to extract object trajectories and train our model with noisy object trajectories and weak supervision labels. A uniform set of hyperparameters for the SAM2.1 mask generator was determined through grid search, and mask quality was manually verified. On average, we extract 19.87 object trajectories for each video, and on average 10.61 objects occur on a single frame. Compared to the OpenPVSG dataset, there are 20.47 object trajectories for each video, and on average 10.36 objects occur on a single frame. For the Action Genome and VidVRD datasets, only coarse-grained bounding boxes are available. During evaluation, these bounding boxes are converted into masks by setting all parts within the bounding boxes to True and areas outside to False.

The chart below illustrates the learning performance of our method, leveraging CLIP as the backbone. The results demonstrate that our approach is robust in handling complex caption descriptions, noisy object trajectories and out-of-domain transfer scenarios.

| Eval Dataset & LASER FT Strategy | | Unary R@K | | | Binary R@K | | |
|---|---|---|---|---|---|---|---|
| Dataset | Strategy | R@1 | R@5 | R@10 | R@1 | R@5 | R@10 |
| OpenPVSG | Base | 0.1633 | 0.3381 | 0.4404 | 0.0197 | 0.0673 | 0.0988 |
| | LLaVA-FT | 0.2368 | 0.5000 | 0.5789 | 0.1191 | 0.3534 | 0.5346 |
| | OpenPVSG-FT | 0.2778 | 0.5231 | 0.6402 | 0.1482 | 0.4214 | 0.5398 |
| Action Gnome | Base | 0.1487 | 0.4166 | 0.5911 | 0.0509 | 0.2464 | 0.5478 |
| | LLaVA-FT | 0.1768 | 0.4795 | 0.6471 | 0.1255 | 0.5250 | 0.6772 |
| | OpenPVSG-FT | 0.1422 | 0.4367 | 0.6277 | 0.2407 | 0.3085 | 0.6927 |
| VidVRD | Base | 0.6267 | 0.8408 | 0.9332 | 0.0499 | 0.1696 | 0.2445 |
| | LLaVA-FT | 0.6444 | 0.8899 | 0.9585 | 0.0678 | 0.1979 | 0.2966 |
| | OpenPVSG-FT | 0.5925 | 0.8322 | 0.9161 | 0.0211 | 0.1179 | 0.1715 |

Table 6: Performance comparison of LASER finetuning strategies across different evaluation datasets. Note that the LLaVA-FT is out of distribution and OpenPVSG-FT is in domain.

#### B.4.1 Using STSL as STSG

We conducted an experiment on the OpenPVSG eval split, comparing the performance of a model directly using the generated STSL data (referred to as STSL-as-STSG) against our LASER fine-tuned CLIP-based STSG model. Here is the detailed experimental setup, partially addressing the aforementioned challenges:

Vocabulary Alignment: To address challenge (a), we use the spaCy package to project caption-extracted STSL keywords into the dataset-specific vocabulary. When a direct match was not found, we took the top-10 predictions based on similarity scores and assigned the similarity score as the predicted likelihood. Assuming Perfect Grounding: To address challenge (b) and (c), we assumed a hypothetical, ideal algorithm that grounds entities and their relations into specific frames of the video perfectly, bypassing errors related to spatial-temporal grounding. Evaluation Metric An object is counted as a match for unary if its name appears in the top-k most likely predictions of the projected STSL within its predicted event duration. Similarly, a pair of objects is counted as a match for binary if their binary relation is among the top-k most likely predictions of the projected STSL within the predicted event duration. Even under these over-optimistic assumptions, our experiment revealed that a model directly using STSL data without fine-tuning performs significantly worse compared to the fine-tuned CLIP-based STSG model. Through our investigation, we found that the performance degradation is primarily due to (d) mismatched granularity, which cannot be easily addressed without fine-tuning. As a side note, our fine-tuned STSG model produces results much faster than STSL-as-STSG during inference time. On average, calling the full STSL-as-STSG pipeline takes about 34.5 seconds per video for extracting the STSL from the video, while directly calling the fine-tuned CLIP model takes about 2.6 seconds per video on the OpenPVSG dataset.

#### B.4.2 State-of-the-art Fully Supervised Baselines

While STTran and TRACE demonstrate strong performance in video scene graph generation, they are closed-domain models with fixed classification layers, limiting their ability to transfer knowledge across datasets.

| Eval Dataset & LASER FT Strategy | | Unary R@K | | | Binary R@K | | |
|---|---|---|---|---|---|---|---|
| Dataset | Strategy | R@1 | R@5 | R@10 | R@1 | R@5 | R@10 |
| OpenPVSG | Base | 0.1633 | 0.3381 | 0.4404 | 0.0197 | 0.0673 | 0.0988 |
| | LLaVA-FT | 0.2368 | 0.5000 | 0.5789 | 0.1191 | 0.3534 | 0.5346 |
| | OpenPVSG-FT | 0.2778 | 0.5231 | 0.6402 | 0.1482 | 0.4214 | 0.5398 |
| | (NEW) STSL-as-STSG | 0.0514 | 0.1132 | 0.1288 | 0.0362 | 0.0583 | 0.0716 |

Table 7: Performance comparison of LASER FT CLIP strategies on the OpenPVSG dataset.

Additionally, their architectures are not designed to handle noisy labels directly extracted from captions, making them less robust in such scenarios.

It is important to note that the primary contributions of STTran and TRACE lie in their carefully designed model architectures for capturing visual features. In contrast, our work focuses on an orthogonal problem: developing a model-agnostic weak supervision training pipeline. Our approach is not in competition with STTran or TRACE but rather complements them. In fact, STTran and TRACE can be adopted as backbone models within our LASER framework, demonstrating that our method enhances their capabilities rather than being mutually exclusive.

Nevertheless, we report unary and binary predicate recall for STTran using the author-provided checkpoint. Note that STTran is trained in-domain on the Action Genome dataset in a fully supervised manner, while our models are trained on out-of-domain datasets using a weakly supervised approach.

| Eval Dataset & Strategy | | Unary R@K | | | Binary R@K | | |
|---|---|---|---|---|---|---|---|
| Dataset | Strategy | R@1 | R@5 | R@10 | R@1 | R@5 | R@10 |
| Action Gnome | Base CLIP | 0.1487 | 0.4166 | 0.5911 | 0.0509 | 0.2464 | 0.5478 |
| | LLaVA-FT | 0.1768 | 0.4795 | 0.6471 | 0.1255 | 0.5250 | 0.6772 |
| | OpenPVSG-FT | 0.1422 | 0.4367 | 0.6277 | 0.2407 | 0.3085 | 0.6927 |
| | STTran | 0.0671 | 0.2879 | 0.4632 | 0.4097 | 0.7815 | 0.8886 |

Table 8: Performance comparison of different strategies on the Action Gnome dataset. We note that LLaVA-FT and OpenPVSG-FT is out of distribution, and weakly supervised, while STTran is in domain and fully supervised.

### B.5   MUGEN

| Task | SDSC | LASER-P |
|---|---|---|
| Video-to-Text Retrieval | 87.00% | **93.80%** |
| Text-to-Video Retrieval | 86.80% | **90.00%** |

Table 9: Comparison to baselines on downstream retrieval tasks with the MUGEN dataset.

**Model Architecture** We present the model architecture overview in Figure 10b. The video is first segmented into clips through a sliding window of length 2. Then, each clip is encoded through an S3D model and yields clip-wise embedding. The embedding is further passed into a BiLSTM model to obtain the context for each clip. An action classifier takes in the concatenation of the clip-based embedding and its context embedding, and classifies each clip into 6 actions: `walk`, `jump`, `kill`, `collect`, `die`, and `climb`. Further, a modifier classifier predicts the 4 possible direction of the actions: `left`, `right`, `up`, `down`. The matching process of action and its modifier is also performed in the reasoner. For example, a combination of `collect`, `left` will be invalidated during reasoning.

**Learning Setup** We train the model on 5000 training datapoints, and 12, 851 test datapoints. The contrastive loss is obtained over a batch size 3; the violation loss is constructed with regard to the axioms in GPA Migimatsu & Bohg (2022). We train the model with a learning rate of 0.0001, violation weight 0.01, number of epochs 100, and batch size 3. The Scallop reasoning engine setup is using difftopkproofs provenance with $k$ set to 5.

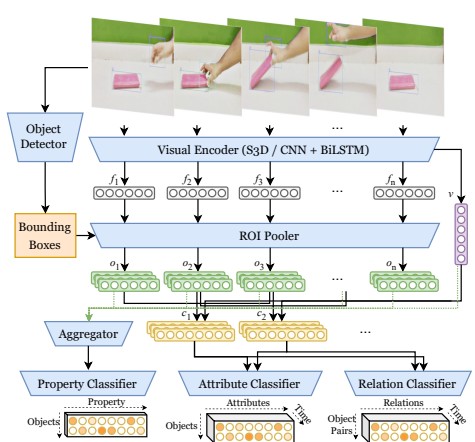

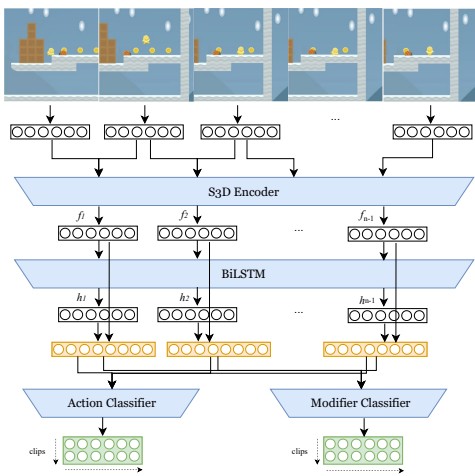

(a) 20BN-Something-Something classifier model architecture used in LASER-P.

(b) MUGEN model classifier architecture used in LASER-P.

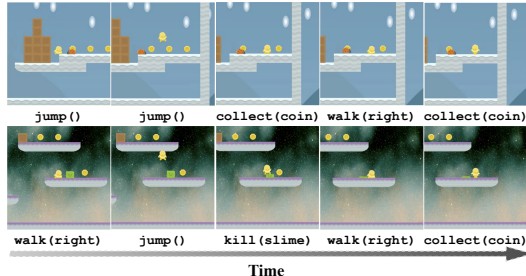

Figure 11: Examples of the predicted MUGEN actions predicted. We elide showing `M` in each predicate for brievity.

We construct the temporal specification from the given text description using a simple heuristic-based semantic parser. We first extract an ordered action list $[a_1, a_2, \ldots, a_n]$ from the text with heuristics, then construct the temporal specification as $a_1 \mathbf{U} a_2 \ldots \mathbf{U} a_n$. This specification means the actions are performed one by one with no intermediate gaps. The generated specifications are used as ground truth programmatic STSL labels. As the MUGEN video data is very different from the natural videos, we train a classification model from scratch. Our neural perception model consists of an S3D video encoding model pretrained on Kinetics 400 Kay et al. (2017), and MLP layers to classify the actions. The video is first passed through the S3D Xie et al. (2018) model for frame-based embeddings. Then all the embeddings that occur in a video clip are concatenated as the input to a 2-layer MLP classifier. We set the number of frames per clip to 2, and the batch size to 3 due to the hardware limitation. The classifier thus produces clip-wise video representations. We employ contrastive loss and semantic loss for training. More details are shown in Appendix B.5.

**Quantitative Study.** We evaluate our action prediction performance on MUGEN and compare it to caption supervised VT-TWINS Ko et al. (2022), TempCLR Yang et al. (2023b), and a directly supervised baseline (Supervised). We provide all baselines with extra annotations on the start- and end-frame of each clip-based textual description. As the ground truth actions for the video are different from the text description, we use heuristics to obtain the start and end frames. For a fair comparison, we fine-tune an S3D backbone pre-trained using Kinetics 400 (the same as LASER) along with the baseline models. This label is used for training both VT-TWINS and fully supervised models. As shown in Table 4, LASER has better action prediction accuracy despite receiving less supervision than the baselines. Moreover, our weakly supervised model even achieves better accuracy than the Supervised method on #Data = 5000. We further evaluate our approach on a downstream video-specification retrieval task. Given 3 videos and 3 specifications, we want to match the correctly aligned pairs, retrieve specification given video and vice versa. We denote the two tasks as spec-retrieval and video-retrieval respectively. Specifically, we evaluate the accuracy with which we infer whether the correct specification has the highest alignment score, and vice versa. LASER outperforms an embedding-based baseline SDSC Hayes et al. (2022) on both tasks (Table 9). Our approach can even identify actions that persist for a very short period of time, such as `kill` an enemy.

## B.6   20BN-Something-Something.

| Frame | 0 | 10 | 20 | 30 |
|---|---|---|---|---|
| **Video: pull right** | | | | |
| **temporal** | 1.000: left(A)
1.000: right(A) | 1.000: left(A)
1.000: right(A) | 1.000: left(A)
1.000: right(A) | 1.000: left(A)
1.000: right(A) |
| **temporal + contrastive** | 0.950: left(A)
0.318: right(A) | 0.732: left(A)
0.484: right(A) | 0.991: left(A)
0.827: right(A) | 0.912: left(A)
0.657: right(A) |
| **temporal + contrastive + violation** | 0.302: left(A)
0.587: right(A) | 0.196: left(A)
0.660: right(A) | 0.236: left(A)
0.819: right(A) | 0.032: left(A)
0.994: right(A) |

Figure 12: Comparing model performance trained with different loss functions.

**Dataset Details** We trained on $10,000$ datapoints and tested on $14,816$ datapoints. There are 172 actions with 37 underlying predicates in this dataset. Specifically, there are 6 static predicates, 21 unary predicates, and 10 binary predicates. The static predicates represent object attributes that persist over time; the unary predicates reflect a single object's state; the binary predicates reflect the relationship between two objects.

**Model Architecture** We use a 4-layer convolutional model to extract the features for each frame and a ROIpooler to obtain the embedding for each object on the frame. To obtain the static predicates, we pass the frame-wise object embeddings through an LSTM encoder to obtain the video-wise object embedding, and a 2-layer MLP classifier yields the output distribution. For the unary predicates, a 2-layer MLP classifier takes in the concatenation of the frame embedding, object embedding, and the object bounding box, and generates the output distribution. For the binary predicates, a 3-layer MLP classifier takes in the concatenation of the frame embedding, two object embeddings, and two object bounding boxes, and generates the output distribution. The overview of the architecture is shown in Figure 2.

We use a convolutional model for the frame features, an ROI Pooler for object features, an LSTM model for video-wise object embeddings, and MLP layers for predicate output distributions. We adopt temporal supervision loss with the prior that the precondition and postcondition should be far away from each other; contrastive learning loss with batch size 3; and semantic loss with a weight of 0.05, where the integrity constraints, such as "a rigid object is not fluid", are obtained from the original PDDL file. More training details are included in the appendix.

Instead of using just querying for the alignment score $Pr(\mathbf{r} \models \psi)$, we instead query for the conditional alignment score $Pr(\mathbf{r} \models \psi | d)$ which is conditioned on the distance $d$ between $\psi_{\text{pre}}$ and $\psi_{\text{post}}$ are satisfied. In the experiment, we set the threshold $d_{\min}$ to be $0.9 d_{\max}$.

**Learning Setup.** The contrastive loss is obtained over a batch size 3; the violation loss is constructed with regard to the axioms in GPA Migimatsu & Bohg (2022). We train the model with a learning rate of $0.0001$, violation weight $0.05$, number of epochs $50$, and batch size 3. The Scallop reasoning engine setup is using difftopkproofs provenance with k=3.

**Ablation Studies.** We study how different loss impact the qualitative evaluation result, as shown in Figure 12. In this task, the spatiotemporal specifications are manually crafted, which also means a lot of biases are introduced. As we can see, using only temporal loss yields us a counter-intuitive result, a 1.00 probability for both the object on the left of the camera and on the right of the camera. This is mainly due to the imbalanced low-level supervision that is introduced by human bias. By adding a contrastive loss, we can see an improvement in that the model predicts the ground truth position with a higher probability compared to its counterpart. Incorporating a violation loss can further improve the performance in that the sum of the probabilities is closer to 1.0.

**More Qualitative Studies** We include more qualitative examples predicted by the LASER-P model in Figure 13.

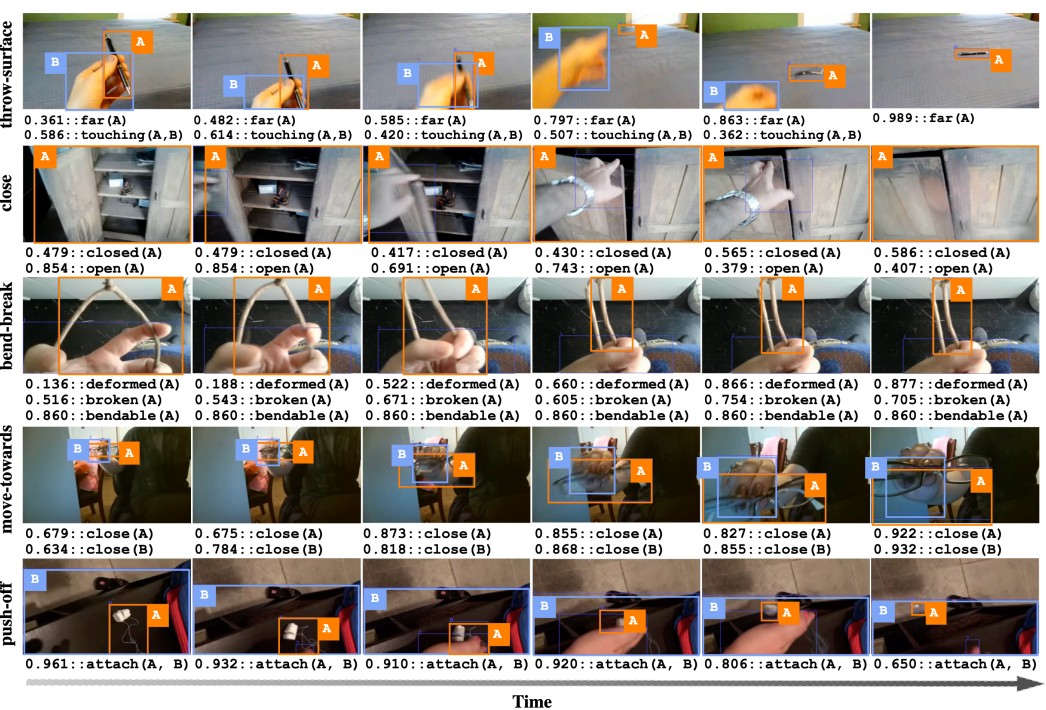

Figure 13: The STSGs predicted by LASER on the 20BN dataset. The actions are labeled on the left.

