# OpenReview forum: "LASER: A Neuro-Symbolic Framework for Learning Spatio-Temporal Scene Graphs with Weak Supervision"
_ICLR.cc/2025/Conference — ICLR 2025 Poster_

### Official Review · Reviewer_ym8p · 2024-10-27

**Soundness:** 3
**Presentation:** 2
**Contribution:** 2
**Rating:** 6
**Confidence:** 2

**Summary:**

The paper presents LASER, a model-agnostic neuro-symbolic framework designed to learn Spatio-Temporal Scene Graphs from videos using weak supervision derived from video captions. Traditional supervised methods for training STSGs are hindered due to reliance on labor-intensive annotated datasets, while LASER addresses this challenge by employing LLMs to extract logical specifications and then training the underlying generator to align the STSG with the specification.

**Strengths:**

1. The overall presentation is clear and the figures are well-illustrated.
2. The introduction of the STSL provides a flexible and expressive way to capture the nuances of video semantics, and the alignment checking ensures the relations between STSL and STSG.
3. Extensive experiments and ablation studies demonstrate the effectiveness of LASER in learning STSG.

**Weaknesses:**

1. The performance of LASER heavily relies on the quality of video captions. If captions are sparse or inaccurate, the resulting generated STSGs may also suffer in quality.
2. The model seems to have a problem building scene graphs with longer videos and more instances.
3.  While the framework is promising, the scalability of the approach to very large datasets or real-time applications may pose challenges to the model.

**Questions:**

1. In Table 3, setup 1 & 2 have the same F1 score, while the full setting of losses is higher than setup 3. Can you explain this phenomenon?
2. See weaknesses.

---

> ### Author Response · Authors · 2024-11-23
>
> ## **Q1:**
> > **"The performance of LASER heavily relies on the quality of video captions. If captions are sparse or inaccurate, the resulting generated STSGs may also suffer in quality."**
>
> ### **Answer:**
> Thank you for pointing out the potential issue that when captions lack sufficient detail, our method may struggle to extract high-quality temporal STSL descriptions. To address this, we can leverage a visual-language model pretrained on video-caption datasets to generate detailed captions for a given video. To alleviate the reviewer’s concern, we have already conducted experiments using LLaVA-Video [1], which introduces a prompting technique to synthesize such detailed captions effectively with GPT-4o. The results are very promising and can be found in the overall rebuttal section under *“Training with Noisy Object Trajectories and Long Captions on LLaVA-Video.”*
>
> [1] Zhang, Yuanhan, et al. "Video Instruction Tuning With Synthetic Data." *arXiv preprint arXiv:2410.02713 (2024).*
>
>
> ## **Q2:**
> > **"The model seems to have a problem building scene graphs with longer videos and more instances.  ""**
>
> > **"While the framework is promising, the scalability of the approach to very large datasets or real-time applications may pose challenges to the model."**
>
> ### **Answer:**
> To address scalability concerns, we include an experiment with the LLaVA-Video dataset, which contains significantly more instances and richer temporal information. This experiment is detailed in the overall rebuttal section under *“Extra Experimental Results: Training on the 10K LLaVA-Video Dataset Showing Generalizability, Robustness, and Scalability.”* Furthermore, we have optimized the training process by incorporating the **DistributedDataParallel module**, which enhances training efficiency on large-scale datasets.
>
> It is also important to note that our LASER pipeline impacts only the **training phase**. During evaluation, the trained model is directly deployed to recognize STSGs, and the test-time efficiency is entirely dependent on the backbone STSG recognition model. If the backbone model supports real-time inference, the LASER fine-tuned model will retain this real-time capability.
>
>
> ## **Q3:**
> > **"In Table 3, Setup 1 & 2 have the same F1 score, while the full setting of losses is higher than Setup 3. Can you explain this phenomenon?"**
>
> ### **Answer:**
> In general, all three loss terms contribute significantly, although their individual contributions may differ across datasets. The primary reason for the observed differences lies in the inherent distinctions of the datasets, including variations in video content, caption content, and temporal characteristics.
>
> In the **20BN dataset**, temporal loss is the most critical factor. The videos in this dataset capture specific actions, with specifications describing the **pre- and post-condition** of these actions. Naturally:
> - The precondition should occur in the early stages of the video.
> - The postcondition should occur in the late stages.
>
> Without the guidance of **temporal loss**, the alignment checker compares the specification against the entire probabilistic STSG, retaining only the top five likely traces. When training from scratch with only the contrastive and semantic losses, the model struggles to converge on appropriate matches for early video frames corresponding to the precondition and late video frames corresponding to the postcondition. Therefore, incorporating temporal loss is crucial for LASER to achieve strong performance on the 20BN dataset, with further improvements achieved by adding contrastive and semantic losses.
>
> As the temporal loss is **necessary for learning on the 20BN dataset**, and both Setup 1 and Setup 2 do not include temporal loss, it explains the relatively low performance observed in these setups. We have also included a qualitative study on the contribution of each loss term for 20BN in **Figure 12 in Appendix B.4**, which provides an intuitive understanding of each loss's impact.

---

> > ### Author Response · Authors · 2024-11-25
> >
> > We appreciate your valuable comments on our paper. We have prepared a rebuttal with an updated manuscript and tried our best to address your concerns. We notice that the author-reviewer discussion period is coming to an end, and we are willing to answer any unresolved or further questions that you may have regarding our rebuttal if time is allowed.
> > If our rebuttal has addressed your concerns, we would appreciate it if you would let us know if your final thoughts. Additionally, we will be happy to answer any further questions regarding the paper. Thank you for your time and consideration.

---

> > > ### Comment · Reviewer_ym8p · 2024-11-26
> > >
> > > The authors have provided answers to most of my concerns. Therefore, I will increase my rating.

---

### Official Review · Reviewer_nqrt · 2024-11-02

**Soundness:** 4
**Presentation:** 3
**Contribution:** 2
**Rating:** 6
**Confidence:** 3

**Summary:**

This paper uses large language models to generate pseudo-labels for spatiotemporal scene graph generation models without intensive human annotations. It generally contains a probabilistic relational database construction section, a spatiotemporal specification language description section, a natural language parsing section, and a spatiotemporal alignment checking section. A loss function that combines contrastive, temporal, and semantic components is exploited. The authors conduct extensive experiments on three popular datasets: OpenPVSG, 20BN, and MUGEN.

**Strengths:**

1. Exploring large language models to generate pseudo labels for vision-language learning tasks is intuitive nowadays. The proposed method designs an elegant pipeline to generate pseudo labels for the spatio-temporal scene graph generation task.

**Weaknesses:**

1. The significance of the target task spatiotemporal scene graphs needs to be discussed further. Since the large language models generate the pseudo label, they should be able to create scene graphs with appropriate prompts. So, the problem formulation and significance do not seem solid enough. It would be better to compare the proposed approach more explicitly to directly using LLMs to generate scene graphs. If the experiments are too expensive, the authors can discuss the potential advantages of the proposed method over state-of-the-art MLLMs. The authors can also explain why training a separate STSG model is beneficial.
2. Some technical details are not clear enough. The symbols in equation (1) are not explained well.

 $~~~~~~~~~~~~~~~~~\mathcal{L}\left ( Pr\left ( M_{\theta} \left ( X \right ) |= \psi  \right ) ,1 \right )  $

I am unclear about what the $\psi$ stands for and why $ M_{\theta} \left ( X \right )$ needs to condition on 1. What does the "1" stand for?

3. There are many key concepts and methodologies that are not explained in the method section. For example, in line 181, what does "0.05::unary_atom("deformed", 3, e)" stand for, especially the number "0.05"? The same problem appears in line 182.

   In lines 213-215, the logic component set is not given. It's an essential point for readers unfamiliar with the spatiotemporal scene graph to understand the paper.

4. The spatio-temporal scene graph part in the right-up corner of Figure 1 is unclear. The same problem appears in Figure 6.

In Figure 1, in the third row, there are "$\neg $touching (B, H)" both on the left and the right side. What this means seems to be ambiguous.

5. The paper does not present state-of-the-art MLLM results. To demonstrate the advantage of the proposed method, authors should consider some multimodal large language models, such as BLIP-2, InstructBLIP, MiniGPT-4, and LLaVA-1.5.

6. The title uses the keyword "neuro-symbolic," but it seems that the proposed method doesn't emphasize this point. It deserves further discussion.

**Questions:**

I have no further questions. See the weakness part please.

---

> ### Author Response · Authors · 2024-11-23
>
> ## **Q1**:
> > **"The significance of the target task spatiotemporal scene graphs needs to be discussed further. ... It would be better to compare the proposed approach more explicitly to directly using LLMs to generate scene graphs. If the experiments are too expensive, the authors can discuss the potential advantages of the proposed method over state-of-the-art MLLMs. The authors can also explain why training a separate STSG model is beneficial."**
>
> > **"The paper does not present state-of-the-art MLLM results. To demonstrate the advantage of the proposed method, authors should consider some multimodal large language models, such as BLIP-2, InstructBLIP, MiniGPT-4, and LLaVA-1.5.  "**
>
> ### **Answer:**
> In predicting video scene graphs, our observations are that MLLMs face significant challenges in localizing textual descriptions to specific parts of images and videos due to the lack of precise labels. In practice, when attempting to leverage MLLMs to predict video scene graphs, we found that prompting them often yields suboptimal results. For example:
> - **VideoLLaMA** [3] frequently produces mismatched outputs, such as incorrect object colors or inaccurate spatial locations.
> - While **GPT-4o** has not yet released its video-language API, our experiments with image prompts revealed difficulties in extracting precise relationships between objects, such as determining which object a person is looking at.
>
> We will include a qualitative analysis of these failure cases in the revised paper to further illustrate existing MLLM limitations.
>
> Additionally, a major advantage of training a separate STSG model instead of invoking the MLLM is **speed**. During inference, we call the STSG backbone model directly. On average:
> - Calling the full MLLM pipeline takes about **34.5 seconds per video** for extracting STSL from the video.
> - Directly calling the CLIP model takes only about **2.6 seconds per video** on the OpenPVSG dataset.
>
> [3] Zhang, Hang, Xin Li, and Lidong Bing. "Video-llama: An instruction-tuned audio-visual language model for video understanding." arXiv preprint arXiv:2306.02858 (2023).
>
> ## **Q2:**
> > **"The title uses the keyword "neuro-symbolic," but it seems that the proposed method doesn't emphasize this point. It deserves further discussion."**
>
> > **"Some technical details are not clear enough. The symbols in Equation (1) are not explained well. I am unclear about what the $ψ$ stands for and why $M_θ(X)$ needs to be conditioned on 1. What does the "1" stand for?
> There are many key concepts and methodologies that are not explained in the method section."**
>
> ### **Answer:**
> To understand Equation (1), it is important to clarify the symbol $⊨$ (*models*), which denotes **semantic implication**. The notation $A ⊨ B$ means that $B$ is true in every model where $A$ is true.
>
> Here:
> - $M_θ(X)$ represents the predicted probabilistic STSG.
> - $ψ$ denotes the STSL specification.
>
> Thus, $Mθ(X) ⊨ ψ$ implies that the relationships, actions, and temporal constraints specified in $ψ$ are consistent with the probabilistic trajectories and interactions predicted by $M_θ(X)$. Consequently, $Pr(Mθ(X) ⊨ ψ)$ represents the **alignment score** between the predicted STSG and the specification $ψ$.
>
> A detailed example of alignment score calculation and explanation for neural-symbolic components is provided in the overall rebuttal under *“Clarifying Symbolic Components and Their Role in Neural-Symbolic Programming”*.
>
>
> ## **Q3:**
> > **"The spatiotemporal scene graph part in the right-up corner of Figure 1 is unclear. The same problem appears in Figure 6. In Figure 1, in the third row, there are "¬touching (B, H)" both on the left and the right side. What this means seems to be ambiguous."**
>
> ### **Answer:**
> In Figure 1, the **x-axis represents time**, while the **color indicates the likelihood** of a given description being true. For example:
> - The third row shows the probability progression for `touching(B, H)`. Initially, `touching(B, H)` has a **low probability**, suggesting a high likelihood for its negation `¬touching(B, H)`.
> - Over time, the probability of `touching(B, H)` increases before gradually decreasing again, returning to a low likelihood.
>
> We will enhance the visual representation of this figure in the revised version to make these concepts clearer.

---

> > ### Author Response · Authors · 2024-11-25
> >
> > We appreciate your valuable comments on our paper. We have prepared a rebuttal with an updated manuscript and tried our best to address your concerns. We notice that the author-reviewer discussion period is coming to an end, and we are willing to answer any unresolved or further questions that you may have regarding our rebuttal if time is allowed.
> > If our rebuttal has addressed your concerns, we would appreciate it if you would let us know if your final thoughts. Additionally, we will be happy to answer any further questions regarding the paper. Thank you for your time and consideration.

---

> > > ### Comment · Reviewer_nqrt · 2024-11-26
> > > **Relpy to the response**
> > >
> > > I have no further questions and tend to maintain the rating. The following versions can enhance the writing details.

---

### Official Review · Reviewer_sYch · 2024-11-04

**Soundness:** 2
**Presentation:** 2
**Contribution:** 3
**Rating:** 6
**Confidence:** 4

**Summary:**

The paper proposes LASER, a framework for training spatio-temporal scene graph (STSG) generators using video captions as weak supervision. LASER uses large language models to extract logical specifications from captions and aligns these with predicted STSGs through a differentiable symbolic reasoner and a mix of contrastive, temporal, and semantic losses. This approach enables training STSG generators without manually annotated STSG data. LASER is evaluated on OpenPVSG, 20BN, and MUGEN datasets, showing notable improvements over supervised baselines.

**Strengths:**

The paper introduces a novel approach for training spatio-temporal scene graph (STSG) models using video captions as weak supervision, which removes the need for labor-intensive STSG annotations. It leverages large language models to extract logical specifications, which is innovative in combining neuro-symbolic learning with weakly supervised STSG generation.

This work has high significance as it opens a new path for STSG learning, making it accessible without costly annotation processes. Its performance gains on OpenPVSG, 20BN, and MUGEN suggest practical implications for scalable STSG generation in video understanding.

**Weaknesses:**

The reliance on large language models to extract logical specifications may introduce noise and inaccuracies, especially if the captions lack detailed descriptions. Enhancing this extraction process or adding error-handling mechanisms could improve overall model performance.

The evaluation covers three datasets but could be strengthened by testing on additional complex datasets for scene graph generation to validate generalizability, such as Action Genome, VidVRD, and VIdOR.

The paper does not include strong works on scene-graph generation, such as STTran, TRACE. It is hard to evaluate the robustness of the proposed framework without the comparison with such methods.

STTran: https://openaccess.thecvf.com/content/ICCV2021/papers/Cong_Spatial-Temporal_Transformer_for_Dynamic_Scene_Graph_Generation_ICCV_2021_paper.pdf

TRACE: https://openaccess.thecvf.com/content/ICCV2021/papers/Teng_Target_Adaptive_Context_Aggregation_for_Video_Scene_Graph_Generation_ICCV_2021_paper.pdf

**Questions:**

Could you clarify the specific criteria or methods used by the language model to extract logical specifications? Additional details on handling ambiguous or underspecified captions would be helpful.

Have you considered using additional forms of weak supervision, such as visual cues or scene descriptions, to compensate for the limitations of captions alone?

How well does the approach generalize to more complex, real-world videos beyond the tested datasets? Testing on diverse, large-scale datasets could provide valuable insights.

---

> ### Author Response · Authors · 2024-11-23
>
> ## **Q1:**
> > **"The reliance on large language models to extract logical specifications may introduce noise and inaccuracies, especially if the captions lack detailed descriptions. Enhancing this extraction process or adding error-handling mechanisms could improve overall model performance."**
>
> > **"Could you clarify the specific criteria or methods used by the language model to extract logical specifications? Additional details on handling ambiguous or underspecified captions would be helpful.""**
>
> ### **Answer:**
> Thank you for pointing out the potential issue that when captions lack sufficient detail, our method may struggle to extract high-quality temporal STSL descriptions. To address this, we can leverage a visual-language model pretrained on video-caption datasets to generate detailed captions for a given video. To alleviate the reviewer’s concern, we have already conducted experiments using LLaVA-Video [2], which introduces a prompting technique to synthesize such detailed captions effectively with GPT-4o. The results are very promising and can be found in the overall rebuttal section under *“Training with Noisy Object Trajectories and Long Captions on LLaVA-Video.”*
>
> [2] Zhang, Yuanhan, et al. "Video Instruction Tuning With Synthetic Data." arXiv preprint arXiv:2410.02713 (2024).
>
> ## **Q2:**
> > **"The evaluation covers three datasets but could be strengthened by testing on additional complex datasets for scene graph generation to validate generalizability, such as Action Genome, VidVRD, and VidOR."**
>
> > **"How well does the approach generalize to more complex, real-world videos beyond the tested datasets? Testing on diverse, large-scale datasets could provide valuable insights."**
>
> > **"Have you considered using additional forms of weak supervision, such as visual cues or scene descriptions, to compensate for the limitations of captions alone?"**
>
> ### **Answer:**
>
> In the overall rebuttal section under *“Training with Noisy Object Trajectories and Long Captions on LLaVA-Video”*, we demonstrate that LASER can fine-tune its backbone model using the long and detailed captions, as well as noisy object trajectories, provided by the LLaVA-Video dataset. These detailed captions can be viewed as comprehensive **scene descriptions**, further enriching the training process. Moreover, LASER showcases strong generalizability across various downstream evaluation datasets, including Action Genome and VidVRD. It is worth noting that VidOR is already included as a subset of the OpenPVSG dataset.
>
> ## **Q3:**
> > **"The paper does not include strong works on scene-graph generation, such as STTran, TRACE. It is hard to evaluate the robustness of the proposed framework without the comparison with such methods."**
>
> ### **Answer:**
> While STTran and TRACE demonstrate strong performance in video scene graph generation, they are closed-domain models with fixed classification layers, limiting their ability to transfer knowledge across datasets. Additionally, their architectures are not designed to handle noisy labels directly extracted from captions, making them less robust in such scenarios.
>
> It is important to note that the primary contributions of STTran and TRACE lie in their carefully designed model architectures for capturing visual features. In contrast, our work focuses on an orthogonal problem: developing a model-agnostic weak supervision training pipeline. Our approach is not in competition with STTran or TRACE but rather complements them. In fact, STTran and TRACE can be adopted as backbone models within our LASER framework, demonstrating that our method enhances their capabilities rather than being mutually exclusive.
>
> Nevertheless, we report unary and binary predicate recall for STTran using the author-provided checkpoint. Note that STTran is trained in-domain on the Action Genome dataset in a fully supervised manner, while our models are trained on out-of-domain datasets using a weakly supervised approach.
>
> | Eval Dataset  | Strategy                   | Unary R@1 | Unary R@5 | Unary R@10 | Binary R@1 | Binary R@5 | Binary R@10 |
> |---------------|------------------------------------------|-----------|-----------|------------|------------|------------|-------------|
> | Action Gnome  | Base CLIP                                    | 0.1487    | 0.4166    | 0.5911     | 0.0509     | 0.2464     | 0.5478      |
> |               | LLaVA-FT (out of distribution, weakly supervised) | **0.1768**    | **0.4795**    | **0.6471**     | 0.1255     | 0.5250     | 0.6772      |
> |               | OpenPVSG-FT (out of distribution, weakly supervised) | 0.1422    | 0.4367    | 0.6277     | 0.2407     | 0.3085     | 0.6927      |
> |               | STTran (in domain, fully supervised)     | 0.0671    | 0.2879    | 0.4632     | **0.4097**     | **0.7815**     | **0.8886**      |

---

> > ### Comment · Reviewer_sYch · 2024-11-25
> >
> > The authors have provided answers to most of my queries. There were a few points I misunderstood and asked for unfair comparisons (STTran, TRACE). After reading the author's response, I am increasing my score.

---

> > > ### Author Response · Authors · 2024-11-25
> > >
> > > Thank you for acknowledging our responses and for taking the time to carefully consider our clarifications. We’re glad to hear that our explanations addressed your concerns and provided the necessary clarity. We will incorporate your valuable feedback into the revised version of our paper. We are sincerely grateful for your thoughtful feedback and for raising your score.

---

### Official Review · Reviewer_p4rQ · 2024-11-08

**Soundness:** 3
**Presentation:** 3
**Contribution:** 2
**Rating:** 6
**Confidence:** 2

**Summary:**

This paper introduces LASER, a framework that leverages video captions as weak supervision to train spatio-temporal scene graph (STSG) generators, addressing the challenge of costly manual STSG annotations. LASER utilizes large language models to derive spatio-temporal semantic specifications from video captions, then aligns STSG predictions with these specifications using a differentiable symbolic reasoner and specialized loss functions. This approach enables efficient training of STSG models without annotated data, achieving notable improvements over fully-supervised baselines on datasets such as OpenPVSG, 20BN, and MUGEN.

**Strengths:**

1. This paper proposes a new framework the uses specification language, for specifying fine-grained video semantics.
2. The experimental results show good performance compared with fully-supervised one.

**Weaknesses:**

1.  The design of this approach seems puzzling. If the STSG can be generated based on a series of advanced models such as CLIP and LLM, why need to train an additional model to make predictions based on the generated data? What are the problems with directly using the generated data as the results, and what is its performance?
2. The description of the experimental section is also quite unclear. For instance, it is not specified which state-of-the-art (SOTA) methods are used for comparison or whether they represent the most advanced approaches. Additionally, there are no specific references provided, making it difficult to assess the relative effectiveness of the proposed method.
3. The motivation behind STSL is not very clear. Its relationship with STSG is also unclear, as well as why this step is necessary.

**Questions:**

Lines 428-430 describe the baseline; however, it seems that no state-of-the-art (SOTA) methods for relation extraction are mentioned. Does this paper surpass the SOTA methods for image or video relation extraction?

---

> ### Author Response · Authors · 2024-11-23
>
> ## **Q1:**
> > **"The design of this approach seems puzzling. If the STSG can be generated based on a series of advanced models such as CLIP and LLM, why is there a need to train an additional model to make predictions based on the generated data? What are the problems with directly using the generated data as the results, and what is its performance?"**
>
> ### **Answer:**
> Imagine the caption: *“A boy is climbing a tree, and another boy is watching him.”* In STSL, this is expressed as:
> `name(v1, “boy”), name(v2, “boy”), name(v3, “tree”), climbing(v1, v3), watching(v2, v1)`.
>
> Note that STSL differentiates between the two boys using variable names. Without fine-tuning CLIP, the original pretrained CLIP (see Fig. 2) cannot make this distinction. For instance, CLIP will likely assign the same probability to *“first boy watching second boy”* and *“second boy watching first boy.”* This is why a simple combination of CLIP and LLM does not suffice. Using STSL to supervise the fine-tuning of CLIP imparts the capability to distinguish such relationships, enabling our CLIP-based SGG model to make accurate predictions.
>
>
> ## **Q2:**
> > **"The description of the experimental section is also quite unclear. For instance, it is not specified which state-of-the-art (SOTA) methods are used for comparison or whether they represent the most advanced approaches. Additionally, there are no specific references provided, making it difficult to assess the relative effectiveness of the proposed method."**
>
> ### **Answer:**
> To the best of our knowledge, our method is the first framework to utilize weak supervision from captions to learn video scene graphs. As such, there are no direct state-of-the-art (SOTA) baselines available for an apple-to-apple comparison. In the OpenPVSG experiment, we benchmarked our performance against **8 fully supervised baselines** provided with the dataset [1]. Unlike these baselines, which rely on full supervision, our method exclusively uses captions as the learning signal.
>
> [1] Yang, Jingkang, et al. "Panoptic video scene graph generation." *Proceedings of the IEEE/CVF Conference on Computer Vision and Pattern Recognition.* 2023.
>
> ## **Q3:**
> > **"The motivation behind STSL is not very clear. Its relationship with STSG is also unclear, as well as why this step is necessary."**
>
> ### **Answer:**
> The Q1 answer provides an explanation of why STSL is essential.
>
> Further, another key feature of LASER is the **“tracking” of objects along the temporal dimension**. For example, consider the video caption:
> > *“A man swims freestyle with fluid precision. The man pushes off the wall, gliding underwater before breaking the surface to take his first breath.”*
>
> In STSL, this is represented as:
> `Until( Finally(name(v1, “man”), name(v2, “wall”), push_off(v1, v2)), Until( Finally(name(v3, “water”), beneath(v1, v3)), Finally(break(v1, v3))) )`
>
>
> Note that **STSL captures the temporal sequence**.
>
> The CLIP-based SGG model we trained operates similarly to image caption-based methods, predicting object(s) and relationships for each frame. However, it lacks the ability to track objects across time. The STSL and symbolic checker (Section 3.4) address this by projecting the temporal sequence captured in STSL onto the CLIP frame-by-frame predictions. Both the STSL and the symbolic checker are non-trivial and significant contributions of our paper.

---

> > ### Author Response · Authors · 2024-11-25
> >
> > We appreciate your valuable comments on our paper. We have prepared a rebuttal with an updated manuscript and tried our best to address your concerns. We notice that the author-reviewer discussion period is coming to an end, and we are willing to answer any unresolved or further questions that you may have regarding our rebuttal if time is allowed.
> > If our rebuttal has addressed your concerns, we would appreciate it if you would let us know if your final thoughts. Additionally, we will be happy to answer any further questions regarding the paper. Thank you for your time and consideration.

---

### Official Review · Reviewer_gWyP · 2024-11-08

**Soundness:** 3
**Presentation:** 3
**Contribution:** 2
**Rating:** 6
**Confidence:** 4

**Summary:**

This paper introduces a neural-symbolic framework to learn spatio-temporal scene graph generation from the weak supervisory signals provided by video captions. It is achieved by utilizing an LLM to convert a video caption into a programmatic specification and then compute the alignment scores by a pretrained vision-language embedding model. Experiments are conducted to show the effectiveness of the proposed weakly-supervised method in performance.

**Strengths:**

1.This paper is well-written and the organization is reasonable and easy to follow.

2.Experimental results show that the proposed method achieves state-of-the-art performance that can even outperform fully-supervised methods.

3.The introduction of neural-symbolic representations and reasoning sounds interesting and could be a promising direction to explore.

**Weaknesses:**

1.Although the authors claimed that they proposes the first framework to learn weakly-supervised scene graph generation from video caption data, the utilization of image caption data in weakly-supervised scene graph generation has been widely explored [1, 2], and I think this work falls into a straightforward adaptation to the video domain, so this contribution is not that appealing.

2.The idea to incorporate LLMs into scene graph generation has also been investigated by several previous works [3, 4]. This makes the novelty of this framework less significant.

3.For using neural-symbolic techniques, as have been mentioned in the manuscript, it could be hard to accurately transform a video caption into a structured representation like formal program. However, the proposed method heavily relies on the extracted programmatic specification for making predictions and this would possibly become a weakness that reduces the robustness of the method. Moreover, if the video captions become even more complex and longer, I am wondering whether the method will simply fail since the programmatic representations can be very inaccurate in such scenario. It is suggested to add experimental results on the model's sensitivity to the LLM's capability.

4.In terms of performance comparison, the authors claimed that their weakly-supervised framework outperforms fully-supervised frameworks, but I think this is kind of unfair since the complexity and external models like LLMs adopted in this work are much more expensive and stronger than many fully-supervised method. So what would be the comparative result if no LLMs are used in this work? If LLMs are necessary for this method to work, then how much complexity and resources will it cost to use the LLMs?

References

[1] Linguistic Structures As Weak Supervision for Visual Scene Graph Generation, Ye et al.

[2] Learning to Generate Scene Graph from Natural Language Supervision, Zhong et al.

[3] GPT4SGG: Synthesizing Scene Graphs from Holistic and Region-specific Narratives, Chen et al.

[4] LLM4SGG: Large Language Models for Weakly Supervised Scene Graph Generation, Kim et al.

**Questions:**

Please refer to the weaknesses.

---

> ### Author Response · Authors · 2024-11-23
>
> ## **Q1:**
> > **"Although the authors claimed that they propose the first framework to learn weakly-supervised scene graph generation from video caption data, the utilization of image caption data in weakly-supervised scene graph generation has been widely explored [1, 2]. I think this work falls into a straightforward adaptation to the video domain, so this contribution is not that appealing."**
>
> ### **Answer:**
> The main difference between LASER and an image caption-based method is the **“tracking” of objects along the temporal dimension**.
>
> Imagine a video caption:
> > *A man swims freestyle with fluid precision. The man pushes off the wall, gliding underwater before breaking the surface to take his first breath.*
>
> In STSL, this will become:
>
> `Until(Finally(name(v1, “man”), name(v2, “wall”), push_off(v1, v2)) Until(Finally(name(v3, “water”), beneath(v1, v3)), Finally(break(v1, v3))))`
>
> Note that **STSL captures the temporal sequence**.
>
> Now, the CLIP-based SGG that we trained is similar to the image caption-based method you mentioned, predicting, for each frame, the object(s) and relationships. However, it does not understand the **“tracking”**. It is not hard to imagine that the checker (Section 3.4) could take the STSL, which captures the temporal sequence, and project it onto the CLIP frame-by-frame predictions. The STSL and the symbolic checker are non-trivial and significant contributions of our paper.
>
> Nevertheless, at the reviewer’s suggestion, we conducted experiments to compare our approach with [2]. Using the caption processing strategy outlined in [2] on the LLaVA-Video dataset, we obtained **25,685 unique unary keywords** and **2,263 unique binary keywords**, but without any temporal information linking relations to specific video frames. Notably, LASER demonstrates superior performance in extracting high-quality binary relation keywords.
>
> | Dataset         | Method                   | Unary Coverage (%) | Binary Coverage (%) |
> |------------------|--------------------------|---------------------|----------------------|
> | OpenPVSG        | Baseline [2]                 | **92.86**              | 10.53               |
> |                 | Our Method               | 91.27              | **89.47**               |
> | Action Genome   | Baseline                 |**80.56**              | 15.38               |
> |                 | Our Method               | **80.56**              | **69.23**               |
> | VidVRD          | Baseline                 | **88.57**              | 15.90               |
> |                 | Our Method               | 82.86              | **31.82**               |
>
> ## **Q2:**
> > **"The idea to incorporate LLMs into scene graph generation has also been investigated by several previous works [3, 4]. This makes the novelty of this framework less significant."**
>
> ### **Answer:**
> We note that faithfully converting the intermediate representation from a caption into an executable specification that respects the temporal sequence is **nontrivial**. Our compilation design, described in the overall rebuttal section *“Compiling GPT-Generated Intermediate Representations into Executable STSL Specifications”*, takes in the GPT representation and converts it into an STSL program. This process includes lexical analysis, syntactic and semantic analysis, and an error handling and recovery system for generating valid and rich STSL programs.
>
> As shown in the overall rebuttal section *“Extracted STSL Program Quality”*, our compilation process achieves a failure rate of **0.78%** on OpenPVSG and **0%** on LLaVA-Video. Importantly, our STSL program supports open-domain keywords, enabling learning from noisy weak supervision labels, which sets it apart from works like [3, 4] that focus on predicting keywords within a target dataset.
>
> Nevertheless, we followed the reviewer’s suggestion and ran experiments using the method from [4]. We preprocessed long captions using the LLM4SGG [4] prompting technique but found its post-processing script for cleansing GPT-generated results to be highly error-prone and prone to hallucinations, especially in scenarios involving long captions. When replicating their method on the LLaVA-Video dataset, only **3.11%** of the GPT-generated responses could be successfully parsed into a list of relational triplets. In contrast, our parser achieves **100% conversion** of GPT responses into executable STSL programs. Further details are available in the overall rebuttal section *“Clarification on Compiling GPT-Generated Intermediate Representations into Executable STSL Specifications”*.

---

> > ### Author Response · Authors · 2024-11-23
> >
> > ## **Q3:**
> > > **"For using neural-symbolic techniques, as mentioned in the manuscript, it could be hard to accurately transform a video caption into a structured representation like a formal program. However, the proposed method heavily relies on the extracted programmatic specification for making predictions, and this could possibly become a weakness that reduces the robustness of the method. Moreover, if the video captions become even more complex and longer, I am wondering whether the method will simply fail since the programmatic representations can be very inaccurate in such scenarios. It is suggested to add experimental results on the model's sensitivity to the LLM's capability."**
> >
> > ### **ANSWER:**
> > Thank you for your suggestion to include an experiment demonstrating the quality of the extracted programs. In response, we have added a new experiment featuring **longer and more complex captions**, **noisy object trajectories**, and **out-of-distribution evaluation data**. The results demonstrate that LASER effectively enhances the performance of the backbone visual language model across various out-of-domain challenging scenarios. Details can be found in the overall rebuttal section, specifically under *“Training with Noisy Object Trajectories and Long Captions on LLaVA-Video”*.
> >
> > ## **Q4:**
> > > **"In terms of performance comparison, the authors claimed that their weakly-supervised framework outperforms fully-supervised frameworks, but I think this is kind of unfair since the complexity and external models like LLMs adopted in this work are much more expensive and stronger than many fully-supervised method. So what would be the comparative result if no LLMs are used in this work? If LLMs are necessary for this method to work, then how much complexity and resources will it cost to use the LLMs?"**
> >
> > ### **ANSWER:**
> > The integration of LLMs into our framework is driven by their ability to leverage embedded commonsense knowledge, which is often difficult to capture with alternative methods. For example, actions like *"jumping"* are typically brief, whereas *"walking"* tends to span a comparatively longer duration. This embedded reasoning makes LLMs an indispensable component of our framework.
> >
> > Additionally, generating all structured representations with GPT costs approximately **$50** and takes around **15 minutes** using parallelization techniques. The compilation process from structured representations to STSL programs is highly efficient, requiring only about **1 minute** for **10K data points**.

---

> > > ### Author Response · Authors · 2024-11-25
> > >
> > > We appreciate your valuable comments on our paper. We have prepared a rebuttal with an updated manuscript and tried our best to address your concerns. We notice that the author-reviewer discussion period is coming to an end, and we are willing to answer any unresolved or further questions that you may have regarding our rebuttal if time is allowed.
> > > If our rebuttal has addressed your concerns, we would appreciate it if you would let us know if your final thoughts. Additionally, we will be happy to answer any further questions regarding the paper. Thank you for your time and consideration.

---

> > > > ### Comment · Reviewer_gWyP · 2024-11-26
> > > >
> > > > The authors' rebuttal has addressed most of my concerns. Overall, I think neural symbolic might be an interesting and meaningful way to address some complex tasks such as spatio-temporal scene graph generation and reasoning, and this paper can be somehow inspiring to the community in this direction. So I raise my score to 6.

---

> > > > > ### Author Response · Authors · 2024-11-26
> > > > >
> > > > > Thank you for acknowledging our responses and for taking the time to carefully consider our clarifications. We’re glad to hear that our explanations addressed your concerns and provided the necessary clarity. We will incorporate your valuable feedback into the revised version of our paper. We are sincerely grateful for your thoughtful feedback and for raising your score.

---

### Author Response · Authors · 2024-11-23
**Overall Rebuttal**

We sincerely thank all reviewers for their thoughtful feedback and constructive comments on our work. We deeply appreciate the recognition of our innovation in using large language models (LLMs) to weakly supervised STSG learning (sYch, nqrt), the novel neural-symbolic paradigm (gWyP, sYch, nqrt), LASER's strong performance (gWyP, p4rQ), the expressiveness and flexibility of our STSL design (p4rQ, ym8p), the elegance of our pipeline design (nqrt), and the clarity of our writing and comprehensiveness of our experiments (gWyP, p4rQ, ym8p). In the following, we address common concerns shared across multiple reviewers and provide detailed responses to individual feedback.

## **1. Clarifying Symbolic Components and Their Role in Neural-Symbolic Programming (gWyP, p4rQ, nqrt)**

To improve the clarity of our approach, we will include a dedicated background section on neuro-symbolic programming. We further elaborate on our neuro-symbolic component here using the example shown in Figure 6 and described in lines 305–309.

Our neuro-symbolic programming methodology is rooted in the concept of a **probabilistic relational database**, where relations are expressed in the format `<prob>::<predicate_name>(<n-tuple>)`. For example, the relation `0.8::name("o1", "hand")` indicates that the entity `"o1"` has a 0.8 probability of being identified as a "hand." In our framework, each entity corresponds to an object trajectory within the video.

The **STSL program** is represented deterministically in our database, such as `1.0::name(v1, "hand")`, which specifies that the variable `v1` must be grounded to an object identified as a `"hand"`. An alignment between this program and the relationship in the STSG above suggests that `v1` could be `"o1"` with a likelihood of $0.8$.

### **(a) Construction of the Probabilistic Database with Video STSG and STSL Program**

In our framework, probabilities are derived from the outputs of our STSG generation model to construct the video’s probabilistic database, while the extracted STSL program remains deterministic. Using this representation, object trajectory recognition results can be expressed as follows:

- `0.8::name("o1", "hand"); 0.2::name("o1", "sponge")`: `"o1"` is recognized as a "hand" with 0.8 probability and as a "sponge" with 0.2 probability.
- `0.1::name("o2", "hand"); 0.9::name("o2", "sponge")`: `"o2"` is identified as a "hand" with 0.1 probability and as a "sponge" with 0.9 probability.

Similarly, relations are represented probabilistically, e.g., `0.7::touch("o1", "o2")` indicates that `"o1"` is touching `"o2"` with a 0.7 probability. For simplicity, we use shorthand notations like `0.7::touch` in Figure 6, which extends these representations across multiple frames:

- **Frame 1**: `0.7::touch`, `0.3::drop`
- **Frame 2**: `0.6::touch`, `0.4::drop`
- **Frame 3**: `0.2::touch`, `0.8::drop`
- **Frame 4**: `0.1::touch`, `0.9::drop`

This probabilistic relational framework bridges neural outputs with symbolic reasoning, forming the basis of our neuro-symbolic approach.

### **(b) Example of STSL Program Grounding**

The STSL specification we aim to ground into this video STSG database is `"touch U drop"`, meaning the agent performs "touch" followed by "drop." The full STSL program is:

`Until(Finally(touch(v1,v2)), Finally(drop(v1, v2)))`.

To evaluate this, our checker uses a dynamic programming algorithm to determine the top-`k` most likely groundings that satisfy the specification. For example, the top-3 proofs are as follows:

1. The agent performs "touch" at frames 1 and 2, and "drop" at frames 3 and 4, with a probability of: $0.7 * 0.6 * 0.8 * 0.9 = 0.3024$
2. The agent performs "touch" at frame 1, and "drop" at frames 2, 3, and 4, with a probability of: $0.7 * 0.4 * 0.8 * 0.9 = 0.2016$
3. The agent performs "touch" at frames 1, 2, and 3, and "drop" at frame 4, with a probability of: $0.7 * 0.6 * 0.2 * 0.9 = 0.0756$

The aggregate probability of these three proofs is: $0.3024 + 0.2016 + 0.0756 = 0.5796$

We refer to this as the **alignment score** between the video STSG and the STSL program.

In real-world scenarios, logical expressions are often far more complex. To handle such cases, we designed a full **domain-specific language** (shown in Figures 4 and 5) and implemented a **neuro-symbolic alignment checker** (detailed in Appendix C) capable of computing alignment scores for any program expressed in this language.

During training, our goals are to:

1. Maximize the alignment score between a video and its corresponding STSL specification, pushing it towards $1$.
2. Minimize the alignment score between a video and an unmatched STSL specification, pushing it towards $0$.

This approach enables contrastive learning and provides fine-grained feedback for our STSG model.

---

> ### Author Response · Authors · 2024-11-23
>
> ## **2. Clarification on Compiling GPT-Generated Intermediate Representations into Executable STSL Specifications (gWyP, ym8p, sYch, nqrt)**
>
> We will include a comprehensive section on STSL parsing and compilation in our revised paper. Converting GPT-generated intermediate representations into executable STSL specifications is a nontrivial task. Below, we outline our compilation pipeline, which translates the structured representation from GPT into an STSL program for video analysis:
>
> ### **(a) Parsing**
> The parser ingests the GPT-generated intermediate structured representation, provided in JSON format, and constructs an Abstract Syntax Tree (AST). This step includes:
> - Normalizing JSON dictionary keys for consistency.
> - Removing invalid or unrecognized JSON dictionary keys.
> - Constructing and parsing the program using PyParse.
>
> ### **(b) Analyzing**
> The analyzer processes the AST to perform validity checks and necessary transformations, producing a validated AST. Key validation tasks include:
> - Checking the logical correctness of operations.
> - Verifying the sizes of unary and binary tuples.
> - Ensuring the consistency of time descriptions.
>
> ### **(c) Synthesizing**
> The synthesizer uses the validated AST to generate an executable STSL program. This step ensures the output program adheres to the formal syntax and semantics of STSL.
>
> Throughout these steps, robust error-handling and recovery mechanisms are integrated to mitigate the potential effects of hallucinations from the large language model. The pipeline guarantees that the resulting STSL program is executable while preserving the open-world diversity of keywords inherent in the GPT-generated representation.

---

> ### Author Response · Authors · 2024-11-23
>
> ## **3. Extra Experimental Results Trained on 10K LLaVA-Video Dataset Showing Generalizability, Robustness, and Scalability (gWyP, sYch, ym8p)**
>
> We present additional results on the 10K LLaVA-Video [1] dataset, showcasing the generalizability, robustness, and scalability of our approach. We evaluate the quality of extracted STSL programs, coverage of ground-truth keywords across datasets, and the pipeline’s ability to handle domain shifts and complex captions. These findings highlight the flexibility and effectiveness of our method in diverse and noisy real-world scenarios.
>
> ### **(a) OpenPVSG Caption to STSL Program**
> - Captions average **12.69 words**, containing an average of **1.69 events** in the GPT-extracted representations.
> - The failure rate for converting captions into valid executable STSL formulas is **0.78%**.
> - A total of **453 unique unary keywords** and **679 unique binary keywords** were identified, covering:
>   - **92.86%** ground-truth unary and **94.74%** ground-truth binary keywords in the OpenPVSG dataset.
>   - **55.56%** ground-truth unary and **61.54%** ground-truth binary keywords in the Action Genome dataset.
>   - **20%** ground-truth unary and **12.88%** ground-truth binary keywords in the VidVRD dataset.
> - Notably, these closed-world datasets include keywords like *"unsure,"* *"other relation,"* and *"not contacting,"* which rarely occur in natural scenes.
>
> ### **(b) LLaVA-Video 10K Dataset Caption to STSL Program**
> To demonstrate the capability of our LLM + compiler pipeline in handling long and complex caption descriptions, we evaluated it on **LLaVA-Video**, a dataset with detailed captions released on October 4, 2024. From a randomly sampled subset of **10,000 video clips** (each under 30 seconds):
> - Captions average **233.05 words**, and the pipeline extracted **4.03 events** per video on average.
> - The failure rate for converting captions into valid executable STSL formulas was **0%**.
> - A total of **18,458 unique unary keywords** and **4,492 unique binary keywords** were identified, covering:
>   - **91.27%** ground-truth unary and **89.47%** ground-truth binary keywords in the OpenPVSG dataset.
>   - **80.56%** ground-truth unary and **69.23%** ground-truth binary keywords in the Action Genome dataset.
>   - **82.86%** ground-truth unary and **31.82%** ground-truth binary keywords in the VidVRD dataset.
> - Notably, **91.99%** of LLaVA-Video samples originate from YouTube, with the remaining samples sourced from ActivityNet, Charades, Ego4D, NextQA, and YouCook2. This highlights a significant domain shift between the LLaVA-Video dataset and all evaluation datasets.
>
> ### **(c) Keyword Analysis**
> We present the top 10 most frequent unary and binary keywords extracted from captions in both datasets to further illustrate the diversity and robustness of our method.
>  LLaVA-Video Unary | Count | LLaVA-Video Binary | Count | OpenPVSG Unary | Count | OpenPVSG Binary | Count |
> |--------------------|-------|--------------------|-------|----------------|-------|-----------------|-------|
> | women             | 1020  | hold              | 2531  | adult          | 823   | on              | 264   |
> | hand              | 886   | wear              | 2275  | child          | 569   | holding         | 211   |
> | man               | 863   | on                | 1103  | man            | 322   | picking         | 163   |
> | text              | 499   | in                | 811   | ball           | 254   | placing         | 148   |
> | child             | 421   | with              | 725   | I              | 206   | using           | 135   |
> | camera            | 419   | adjust            | 386   | dog            | 187   | toward          | 113   |
> | character         | 414   | color             | 362   | toy            | 127   | in              | 90    |
> | hands             | 256   | stand             | 346   | woman          | 111   | throwing        | 83    |
> | room              | 243   | near              | 286   | Baby           | 106   | playing         | 66    |
> | object            | 239   | sit               | 275   | camera         | 104   | sitting on      | 65    |
>
> ### **(d) Cost of Generating the STSL from Intermediate Representations**
> - Generating all intermediate representations with GPT for the 10K dataset costs approximately **$50** and takes around **15 minutes** using parallelization techniques.
> - The compilation process from the 10K GPT-generated structured representations to executable STSL programs requires about **1 minute**.
>
> [1] Zhang, Yuanhan, et al. "Video Instruction Tuning With Synthetic Data." arXiv preprint arXiv:2410.02713 (2024).

---

> > ### Author Response · Authors · 2024-11-23
> >
> > ### **(e) Training with Noisy Object Trajectories and Long Captions on LLaVA-Video**
> > As the LLaVA-Video dataset does not provide ground truth object mask-level trajectories. To address this, we preprocess the videos using **SAM2.1** to extract object trajectories and train our model with noisy object trajectories and weak supervision labels. A uniform set of hyperparameters for the SAM2.1 mask generator was determined through grid search, and mask quality was manually verified.
> > On average, we extract **19.87** object trajectories for each video, and on average **10.61** objects occur on a single frame. Compared to the OpenPVSG dataset, there are  **20.47** object trajectories for each video, and on average **10.36** objects occur on a single frame.
> > For the Action Genome and VidVRD datasets, only **coarse-grained bounding boxes** are available. During evaluation, these bounding boxes are converted into masks by setting all parts within the bounding boxes to `True` and areas outside to `False`.
> >
> > The chart below illustrates the learning performance of our method, leveraging **CLIP** as the backbone. The results demonstrate that our approach is robust in handling complex caption descriptions, noisy object trajectories and out-of-domain transfer scenarios.
> >
> >
> > | Eval Dataset | LASER FT CLIP Strategy  | Unary R@1 | Unary R@5 | Unary R@10 | Binary R@1 | Binary R@5 | Binary R@10 |
> > |--------------|--------------------------|-----------|-----------|------------|------------|------------|------------|
> > | OpenPVSG     | Base                    | 0.1633    | 0.3381    | 0.4404     | 0.0197     | 0.0673     | 0.0988     |
> > |              | LLaVA-FT (out of distribution) | 0.2368    | 0.5000    | 0.5789     | 0.1191     | 0.3534     | 0.5346     |
> > |              | OpenPVSG-FT (in domain) | **0.2778**    | **0.5231**    | **0.6402**     | **0.1482**     | **0.4214**    | **0.5398**     |
> > | Action Gnome | Base                    | 0.1487    | 0.4166    | 0.5911     | 0.0509     | 0.2464     | 0.5478     |
> > |              | LLaVA-FT (out of distribution) | **0.1768**    | **0.4795**    | **0.6471**     | 0.1255     | **0.5250**     | 0.6772     |
> > |              | OpenPVSG-FT (out of distribution) | 0.1422    | 0.4367    | 0.6277     | **0.2407**     | 0.3085     | **0.6927**     |
> > | VidVRD       | Base                    | 0.6267    | 0.8408    | 0.9332     | 0.0499     | 0.1696     | 0.2445     |
> > |              | LLaVA-FT (out of distribution) | **0.6444**    | **0.8899**    | **0.9585**     | **0.0678**     | **0.1979**     | **0.2966**     |
> > |              | OpenPVSG-FT (out of distribution) | 0.5925    | 0.8322    | 0.9161     | 0.0211     | 0.1179     | 0.1715     |

---

> > > ### Author Response · Authors · 2024-11-27
> > >
> > > We sincerely thank all the reviewers for your time, effort, and invaluable feedback on our work. Your insightful comments and constructive suggestions have been instrumental in helping us refine and strengthen our research. We are especially grateful for your thoughtful engagement, which has greatly enhanced the clarity and impact of our submission. Thank you once again for your constructive and encouraging feedback.

---

### Meta-Review · Area_Chair_gxYN · 2024-12-20

**Metareview:**

The paper presents LASER, a model-agnostic neuro-symbolic framework designed to learn Spatio-Temporal Scene Graphs from videos using weak supervision derived from video captions. Traditional supervised methods for training STSGs are hindered due to reliance on labor-intensive annotated datasets, while LASER addresses this challenge by employing LLMs to extract logical specifications and then training the underlying generator to align the STSG with the specification. A loss function that combines contrastive, temporal, and semantic components is exploited. LASER is evaluated on OpenPVSG, 20BN, and MUGEN datasets, showing notable improvements over supervised baselines.

Strengths:
+ This paper proposes a new framework that uses specification language, for specifying fine-grained video semantics.
+ The experimental results show good performance compared with a fully-supervised one.
+ This paper is well-written and the organization is reasonable and easy to follow.

Initial Weaknesses:
+ The idea to incorporate LLMs into scene graph generation has also been investigated by several previous works. The novelty of the proposed method is limited.
+ The motivation is not clear, and the design of the approach seems puzzling.
+ The paper does not include strong works on scene-graph generation, such as STTran, or TRACE. It is hard to evaluate the robustness of the proposed framework without the comparison with such methods.
+ More results on state-of-the-art MLLMs are needed.

**Additional Comments On Reviewer Discussion:**

After the rebuttal, most of the initial concerns have been addressed, and all reviewers raised their scores to positive ratings.

---

### Decision · Program_Chairs · 2025-01-22

Accept (Poster)